# Asymmetric small-molecule acceptor enables suppressed electron-vibration coupling and minimized driving force for organic solar cells

Jing Guo [1] ✉, Shucheng Qin [2], Jinyuan Zhang[2], Can Zhu[2], Xinxin Xia[3], Yufei Gong [2], Tongling Liang[4], Yan Zeng[2], Guangchao Han[2], Hongmei Zhuo[2], Yuechen Li [2], Lei Meng [2], Yuanping Yi [2], Jianhui Chen[1] ✉, Xiaojun Li [2] ✉, Beibei Qiu [5] ✉ & Yongfang Li [2,6]

Minimizing the energy loss, particularly the non-radiative energy loss ($\Delta E_{nr}$), without sacrificing the charge collection efficiency, is the key to further improve the photovoltaic performance of organic solar cells (OSCs). Herein, we proposed an asymmetric molecular design strategy, via developing alkyl/thienyl hybrid side chain based asymmetric small molecule acceptors (SMAs) BTP-C11-TBO and BTP-BO-TBO, to manipulate the intermolecular interactions to realize enhanced luminescence efficiency and reduced energy loss. Theoretical and experimental results indicate that compared to the three symmetric SMAs BTP-DC11, BTP-DTBO and BTP-DBO, the asymmetric SMAs BTP-C11-TBO and BTP-BO-TBO exhibit repressed electron-vibration coupling and reduced $\Delta E_{nr}$. Moreover, the asymmetric nature of BTP-BO-TBO allows the formation of multiple D:A interfacial conformations and interfacial energies, which have made the charge-transfer state energies closer to that of the strongly absorbing (and emitting) local-exciton state, thus gaining the low $\Delta E_{nr}$ while maintaining efficient exciton dissociation. Consequently, the PM6:BTP-BO-TBO-based OSCs achieve a higher power conversion efficiency of 19.76%, with a high open circuit voltage of 0.913 V and an efficient fill factor of 81.17%, profiting from the more improved and balanced charge mobility and longer carrier lifetime. This work provides molecular design ideas to suppress non-radiative decay and paves the way to obtain high-performance OSCs.

Organic solar cells (OSCs) have distinct merits such as lightness, thinness, flexibility, translucency, and colorization[1–5], which can meet special applications in multiple scenarios such as the Internet of Things, smart buildings, indoor photovoltaics, and wearable devices[6–9], and have become one of the most promising green energy technologies. Thanks to advances in organic semiconductor photovoltaic materials and device processing techniques, particularly the swift advancement of A-DA'D-A type small-molecule acceptors (SMAs)[10–12], the power conversion efficiency (PCE) has exceeded 19% currently for a single junction OSC[13–15]. However, compared to the more efficient inorganic materials-based photovoltaic technologies, e.g. the devices based on crystalline silicon, GaAs, or perovskite, there

is still considerable room for further improvement in the photovoltaic performance of OSCs due to the relatively higher energy loss ($E_{loss}$) and lower fill factor (FF).

Generally, minimizing the non-radiative voltage loss ($\Delta V_{nr}$) is essential for enhancing the open-circuit voltage ($V_{OC}$) and hence the photovoltaic properties of OSCs. Further innovation and development of organic semiconductor materials have been recognized as the key to solving this issue. Gao et al. demonstrated that a low energy offset between donor and acceptor molecular states and high photoluminescence yield of the low bandgap material are the two key factors for minimizing the voltage losses ($V_{loss}$) of OSCs[16,17]. Recent studies have demonstrated that introducing luminescent moieties such as fused ring groups (thiophene, benzene, etc.) to enhance the rigidity of molecular structures and/or the extent of electron delocalization could be an effective method for achieving highly luminescent donor and acceptor molecules, as well as optimal photovoltaic performance[18,19]. In addition, the film morphology is closely related to the charge transport properties, and thus the FF and overall efficiency of OSCs. Synergistically improving the luminescence efficiencies of materials and optimizing the morphological structure of the active layer have become the key to further improve the photovoltaic properties but with great challenges. Therefore, manipulating the intermolecular interactions between donor molecules, acceptor molecules, and/or donor–acceptor molecules to control the intermolecular coupling properties becomes a more viable strategy to realize suppressed $E_{loss}$ and improved FF.

Inspired by their specific electronic structure and physical properties, modulating the chemical structures of A-DA'D-A type SMAs have been extensively utilized by regulating the three fundamental building blocks, including the central fused ring core unit (DA'D), electron-withdrawing end unit (A), and solubilizing side chains. Owing to the special electron delocalization characteristics, the singlet exciton (SE) excited state of Y-series acceptors could be transferred into the intra-moiety excited (i-EX) state (or namely delocalized singlet exciton (DSE)), which is beneficial for charge generation and decreased non-radiative energy loss[20,21]. Modifications of the side chains, such as the variation of length, topology (linear or branched), branching points, and dimension are the most frequently employed approaches for precisely modulating the solubility, molecular crystallization, and stacking behavior of SMAs[22–24]. For instance, Yan et al. discovered that different positions of alkyl-chain-branching can alter the molecular stacking of SMAs, optimizing phase separation and exciton dissociation[25]. It has been well established that the molecular stacking behavior can be regulated with improved structural order and charge transport in thin films by replacing the linear n-undecyl chain on Y6 with branched 2-butyloctyl, delivering significantly promoted open-circuit voltage ($V_{OC}$) and fill factor (FF)[26]. Additionally, breaking the symmetry of the alkyl chains is also an effective way to boost photovoltaic performance[20,27]. For example, Yang and coworkers developed the hybrid cycloalkyl-alkyl chain-based symmetric/asymmetric acceptors Y-C10ch/A-C10ch, and the PM6:A-C10ch device based on asymmetric molecules had less energy loss[28]. Development of the acceptors with asymmetric side chains, and an in-depth understanding of the intrinsic properties of their molecules, such as molecular packing, electron coupling, and charge transport properties, especially the electron-vibrational coupling, which describes the deformations of the molecular geometries in the course of the electron-transfer process and reflects the interactions between electrons and intramolecular vibrations. Reducing the electron-vibrational coupling has been demonstrated as an effective strategy to suppress the non-radiative recombination, thus leading to suppressed non-radiative energy loss[29]. Therefore, a comprehensive investigation of the molecular performance of side-chain asymmetric acceptors in combination with detailed density functional theory (DFT) horizontal calculations and single-crystal structures is essential for a deeper understanding of the energy loss and charge transport mechanisms of their devices.

To investigate deeply the effects of SMAs with symmetric or asymmetric alkyl-thienyl chains on the device performance from the perspectives of photophysical properties, morphological features, charge dynamic behaviors, etc., two SMAs with asymmetric alkyl/thienyl outer side chains (BTP-C11-TBO and BTP-BO-TBO, respectively) were synthesized in this study and compared in detail with symmetric SMAs (BTP-DC11, BTP-DBO and BTP-DTBO). In comparison to the symmetrical SMAs, the side-chain asymmetrical structure of BTP-C11-TBO and BTP-BO-TBO presented slightly broader optical bandgaps with elevated LUMO energy levels. The molecular packing results examined by GIWAXS indicate that the asymmetric alkyl/thienyl outer side chains endow the acceptors with a more planar skeleton and promote a denser network packing than the corresponding symmetric SMAs, resulting in the higher domain purity in active layers. Furthermore, the asymmetric alkyl/thienyl outer side chains can subtly modulate the D/A interfacial energetics and inhibit molecular vibration, thereby expediting exciton dissociation and mitigating energetic disorder, resulting in enhanced charge generation efficiency and reduced charge recombination. Consequently, these factors contribute to the remarkable photovoltaic performance of the OSCs based on PM6:BTP-BO-TBO and PM6:BTP-C11-TBO, with the maximum PCEs of 19.76% and 18.51%, respectively. Overall, our research not only demonstrated that the asymmetric alkyl/thienyl outer side chain strategy is a feasible approach for reducing non-radiative energy loss, improving charge generation, suppressing charge recombination, and thus boosting the efficiency of OSCs but also provided an instructive guideline for the further development of emerging organic semiconductor materials and organic optoelectronics.

## Results
### Physical properties of the acceptors

Molecular structures of the symmetrical SMAs BTP-DC11, BTP-DBO, and BTP-DTBO and the asymmetric SMAs BTP-C11-TBO and BTP-BO-TBO were shown in Fig. 1a, and their synthetic details are displayed in Supplementary Fig. 1 in Supplementary Information (SI). The ultraviolet-visible (UV-vis) absorption spectra in chloroform (CF) solution state as well as in solid film state are displayed in Fig. 1b, and the related optical parameters are listed in Table 1. In CF solution, the five SMAs exhibit nearly identical absorption spectra, with the coincident maximum absorption wavelength ($\lambda_{max}$) located at about 733 nm, and their absorption coefficients are measured to be in the range of 2.34-2.61×$10^5$ M cm$^{-1}$ (Supplementary Fig. 2 and Supplementary Table 1), indicating that the side chain structures of SMAs exert minimal influence on the intramolecular charge-transfer absorption. In film state, both symmetric and asymmetric SMAs exhibit a similar trend, manifesting notable redshifts when compared with their corresponding solution absorptions (94 nm for BTP-DC11, 77 nm for BTP-DTBO, 68 nm for BTP-DBO, 89 nm for BTP-C11-TBO, 73 nm for BTP-BO-TBO, respectively). In addition, the asymmetric SMAs of BTP-C11-TBO ($1.02 \times 10^5$ cm$^{-1}$) and BTP-BO-TBO ($1.07 \times 10^5$ cm$^{-1}$) present higher absorption coefficients than those of the symmetric SMAs (BTP-DC11, BTP-DTBO, and BTP-DBO), suggesting the distinct aggregation behavior of these five SMAs in the solid state. Meanwhile, according to their absorption edge onsets (the intersection of the tangent line of the absorption peak edge and the vertical axis Y = 0), the optical bandgaps of BTP-DC11, BTP-DTBO, and BTP-DBO, BTP-C11-TBO, and BTP-BO-TBO are estimated to be 1.359, 1.348, 1.379, 1.336, and 1.375 eV, respectively. In both solutions and films, the stokes shifts of BTP-BO-TBO are smaller than that of other SMAs, suggesting that the excited-state relaxation in BTP-BO-TBO is smaller, which is beneficial for the associated voltage losses. This is in agreement with the reduced reorganization energy for the transition between the ground state and the first excited state ($S_1$). In addition, compared with the solutions, the films exhibit relatively larger stokes shifts (Supplementary Fig. 3). This is presumably due to the fact that there exists an energy disorder for the $S_1$ state in the films,

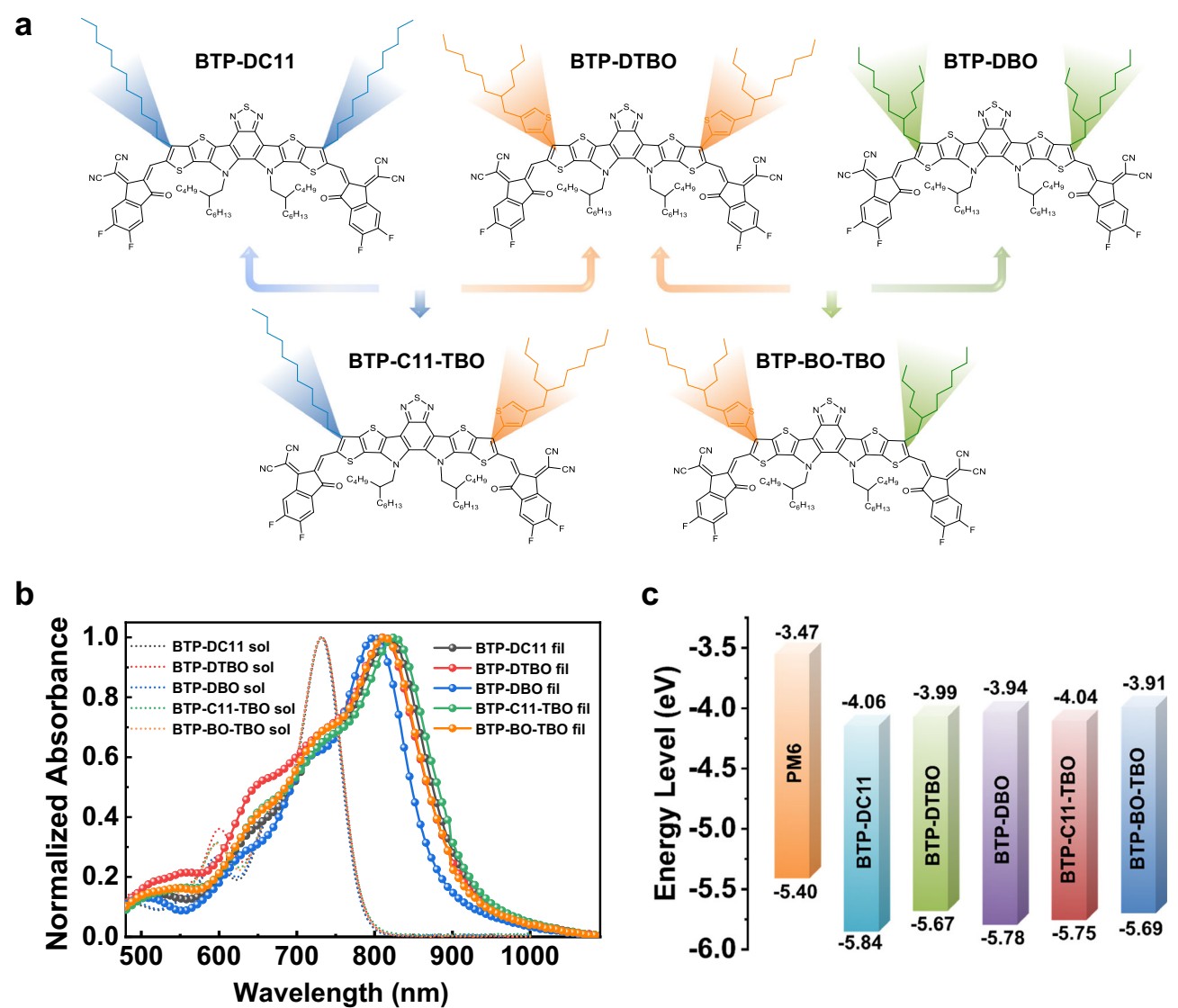

**Fig. 1 | Chemical structures and basic characteristics of the SMAs. a** Molecular structure of BTP-DBO, BTP-DTBO, BTP-DBO, BTP-C11-TBO, and BTP-BO-TBO. **b** UV-vis absorption spectra of solution (dotted lines) and films (solid lines) of five SMAs. **c** Energy-state diagram of PM6 and five SMAs.

**Table 1 | Photophysical and electrochemical properties of SMAs**

| Acceptors | TGA [°C] | $\lambda_{max}$ [nm] | | $\lambda_{onset}$ [nm] | $E_g^{opt}$ [eV][a] | $E_{HOMO}^{CV}$ [eV][b] | $E_{LUMO}^{CV}$ [eV][b] | $E_{HOMO}^{DFT}$ [eV][c] | $E_{LUMO}^{DFT}$ [eV][c] |
|---|---|---|---|---|---|---|---|---|---|
| | | $\lambda_{sol}$ | $\lambda_{film}$ | | | | | | |
| BTP-DC11 | 360.9 | 730 | 824 | 912 | 1.359 | −5.84 | −4.06 | −5.88 | −3.77 |
| BTP-DTBO | 345.7 | 733 | 810 | 920 | 1.348 | −5.67 | −3.99 | −5.88 | −3.74 |
| BTP-DBO | 366.5 | 733 | 801 | 899 | 1.379 | −5.78 | −3.94 | −5.91 | −3.79 |
| BTP-C11-TBO | 360.9 | 732 | 821 | 928 | 1.336 | −5.75 | −4.04 | −5.88 | −3.76 |
| BTP-BO-TBO | 366.1 | 731 | 804 | 902 | 1.375 | −5.69 | −3.91 | −5.90 | −3.77 |

[a]Calculated from the absorption onsets of acceptors; $E_g^{opt} = 1240/\lambda_{onset}$.
[b]Calculated from the onsets of reduction/oxidation potentials.
[c]Calculated the $E_{HOMO}$ and $E_{LUMO}$ by density functional theory (DFT).

and the excitons on the molecules with higher $S_1$ energy can transfer to the molecules with lower $S_1$ energy to emit photons[30]. Generally speaking, the higher the PL fluorescence quantum yield (PLQY), the smaller the non-radiative energy loss, and the results in Supplementary Table 2 show that BTP-C11-TBO and BTP-BO-TBO have the highest PLQY in both solution and thin film, which also fully confirms the above results[31].

The electronic energy levels of five acceptors were measured by electrochemical cyclic voltammetry (CV)[32] (Supplementary Fig. 4). The HOMO/LUMO energy values of BTP-DC11, BTP-DTBO and BTP-DBO, BTP-C11-TBO, and BTP-BO-TBO are calculated to be −5.84/−4.06 eV, −5.67/−3.99 eV, −5.78/−3.94 eV, −5.75/−4.04 eV, and −5.69/−3.91 eV, respectively (Fig. 1c). From the symmetric SMAs BTP-DC11, BTP-DTBO and BTP-DBO to the asymmetric SMAs BTP-C11-TBO and BTP-BO-TBO,

the slightly elevated LUMO level should be attributed to the weak electron-donating properties of thiophene unit, which may contribute to a larger $V_{OC}$ in resulting OSCs. The density functional theory (DFT) calculations were performed at the B3LYP/6-31 G (d,p) level to study the molecular geometry and the frontier molecular orbital energy levels of symmetric SMAs (BTP-DBO, BTP-DTBO, and BTP-DBO) and asymmetric SMAs (BTP-C11-TBO and BTP-BO-TBO). As is demonstrated in Table 1 and Supplementary Fig. 5, the calculated HOMO/LUMO electronic energy levels are −5.88/−3.77 eV for BTP-DC11, −5.88/−3.74 eV for BTP-DTBO, −5.91/−3.79 eV for BTP-DBO, −5.88/−3.76 eV for BTP-C11-TBO, and −5.90/−3.77 eV for BTP-BO-TBO, respectively. Besides, as shown in Supplementary Figs. 6 and 7, the asymmetric structure appears to induce a partial weakening of the molecular configuration in comparison to the symmetrical structure. This results in maintaining dihedral angles and molecular dipole moments within a range intermediate between those of BTP-DC11, BTP-DTBO, and BTP-DBO molecules for both BTP-C11-TBO and BTP-BO-TBO.

Due to the close relationship between the energy loss mechanism and charge transport properties, the impact of asymmetric alkyl/thienyl side chain structure on the electronic structure properties of the acceptors was thoroughly investigated through DFT calculations, the source data are provided in Supplementary Data 1–5. Unlike the charge transport with band transport in inorganic semiconductors such as crystalline silicon, charge transport in thin films composed of organic molecules is generally assumed to occur through a hopping mechanism. Assuming a thermoneutral transport in the thin films, the rate of this charge hopping $k_{ct}$ is given by the Marcus rate equation[33]:

$$k_{ct} = \frac{V^2}{\hbar} \sqrt{\frac{\pi}{\lambda k_B T}} e^{\frac{-\lambda}{4k_B T}} \qquad (1)$$

where the rate depends on the reorganization energy $\lambda$ and the electron coupling between adjacent molecules $V$, whereas the charge recombination energy $\lambda$ dominates. The non-radiative recombination is highly related to electron-vibration coupling (i.e., reorganization energy $\lambda$)[34]. According to the above theory, the smaller the reorganization energy, the smaller the driving force required for exciton dissociation, and the faster the rate of charge transfer and transport. Therefore, the reorganization energy plays an important role in the photoelectric conversion and $E_{loss}$ of OSCs[35].

The reorganization energies between different electronic states in photoelectric conversions are shown in Fig. 2a. The reorganization energy for the $S_0$ to $S_1$ transition ($\lambda_{S0 \to S1}$) is associated with the geometric relaxation of the $S_1$ state after light absorption, while the reorganization energy of the $S_1$ to $S_0$ transition ($\lambda_{S1 \to S0}$) is relevant for the non-radiative exciton decay from the excited state ($S_1$) back to the ground state ($S_0$), and the sum of the two is the reorganization energy for exciton diffusion ($\lambda_{EET}$, exciton energy transfer). The process of exciton dissociation into CT states has two pathways: one is the dissociation of donor exciton by electron transfer, and the other is the dissociation of acceptor exciton by hole transfer, which are responsible for the reorganization energies for the transitions from $S_0$ to anion ($\lambda_{S0 \to Anion}$) and from $S_1$ to anion ($\lambda_{S1 \to Anion}$), and both are related to the reorganization energy for exciton dissociation ($\lambda_{ED}$). The non-radiative charge recombination ($\lambda_{CR}$) hinges on the recombination energy of the anion→S0 transition ($\lambda_{anion \to S0}$)[30].

As shown in Fig. 2b, significantly reduced reorganization energies (below 0.2 eV) between the $S_0$ and $S_1$ transitions were obtained for both the symmetric SMAs system and the asymmetric SMAs system, which indicates that exciton decay is inhibited. The reorganization energies between $S_0$ and $S_1$ to the anion transition in the BTP-C11-TBO and BTP-BO-TBO molecular structures were slightly higher than those of BTP-DC11 and BTP-DBO, and significantly lower than that of BTP-DTBO. It is noteworthy that during exciton dissociation to the CT state, the reorganization energy ($\lambda_{S1 \to anion}$ <0.1 eV) of the hole-transfer

exciton dissociation for all acceptors is very small due to the geometric similarity of the $S_1$ state and the anionic state, which helps to reduce the $V_{loss}$ during exciton dissociation. However, the recombination energy of the exciton dissociated by electron transfer ($\lambda_{S0 \to anion}$) is slightly higher and greater than 0.1 eV. In the process from CT state to charge-separated (CS) state, whether the CT state decays into the ground state or the partially separated free carriers recombine back to the ground state, both processes generally cause energy loss. Interestingly, the reorganization energies for the transitions between the anion and $S_0$ states are all below 0.15 eV for both asymmetric acceptors (BTP-C11-TBO and BTP-BO-TBO) and symmetric acceptors (BTP-DC11, BTP-DTBO, and BTP-DBO), which is conducive to reducing non-radiative recombination loss and accelerating electron transport.

The reorganization energy may be further decomposed into the contribution of each vibrational mode in the acceptor, as shown in Fig. 2c. In the transition from $S_1$ to $S_0$, all the five acceptor molecules have a dominant vibrational mode and correspond to the side chain moiety with a frequency of 1615 cm$^{-1}$, which should be ascribed to the alkyl side chains. Among them, both the undecyl side chain in BTP-DC11 and the 2-butyloctyl side chain in BTP-DBO exhibit relatively strong vibrations (about 22 meV). Interestingly, the vibration intensity is significantly reduced in the SMAs with the thiophene side chain, indicating that the introduction of a two-dimensional thiophene side chain can effectively inhibit the vibration of the backbone chain. The vibration of the asymmetric acceptors (BTP-C11-TBO and BTP-BO-TBO) with both the alkyl chain and thiophene side chain display the lower vibrations below 20 meV, thereby promoting the reduction of exciton decay and facilitating exciton transfer. Similarly, there are three dominant vibrational modes from anion to $S_0$ state, in addition to the tensile vibration of the C-C bond and C-H located at 1320 cm$^{-1}$, the linear alkyl side chain (BTP-DC11) has strong vibration at 1170 cm$^{-1}$, and the characteristic vibration frequency of branched alkyl side chain (BTP-DBO) is 1495 cm$^{-1}$, while the two vibrations are significantly suppressed in the thiophene alkyl side chain substituted SMA (BTP-DTBO). More interestingly, all three vibrational modes are significantly inhibited in both the asymmetric acceptor molecules (BTP-C11-TBO and BTP-BO-TBO), aiming to reduce non-radiative recombination. Ultimately, these results further suggest that asymmetric structures can reduce $E_{loss}$ by inhibiting C-C bond stretching, thereby improving the efficiency of OSCs.

To gain an in-depth understanding of the exciton dissociation and recombination dynamics, the BTP-DC11, BTP-DTBO, BTP-DBO, BTP-C11-TBO, and BTP-BO-TBO-based systems were investigated by femtosecond transient absorption (TAS) spectroscopy, as shown in Fig. 3. The 2D transient absorption spectra and the corresponding transient absorption spectra of these blend films at different decay times are displayed in Supplementary Figs. 8 and 9 in the Supplementary Information, respectively. A pump wavelength of 800 nm was adopted to selectively excite the acceptors of the D:A blends. Figure 3a, b compare the transient dynamics of the ground-state-bleach (GSB) signal of donor probing at 633 nm and excited-state absorption (ESA) signal of acceptor probing at 990 nm in blend films, respectively. This donor GSB feature is an indication of ultrafast hole transfer from the SMA exciton to PM6, producing the CT state at the donor−acceptor interface. For the acceptor ESA signal, in comparison with the symmetrical acceptor-based blends, the BTP-C11-TBO and BTP-BO-TBO-based blends show a relatively higher intensity of ESA signal centered at 900 nm after photoexcitation. Subsequently, as the ESA peak gradually decreased, a redshifted ESA signal on a long timescale from 20 to 1000 ps could be observed, which should be assigned to the polaron states. The stronger and prolonged polaron states of the BTP-C11-TBO and BTP-BO-TBO-based blend films indicated the lower bimolecular recombination probability in the corresponding devices. In addition, for the GSB signal of the donor component, the kinetic curve of the BTP-C11-TBO-based blend reached the maximum value at the beginning and then rapidly

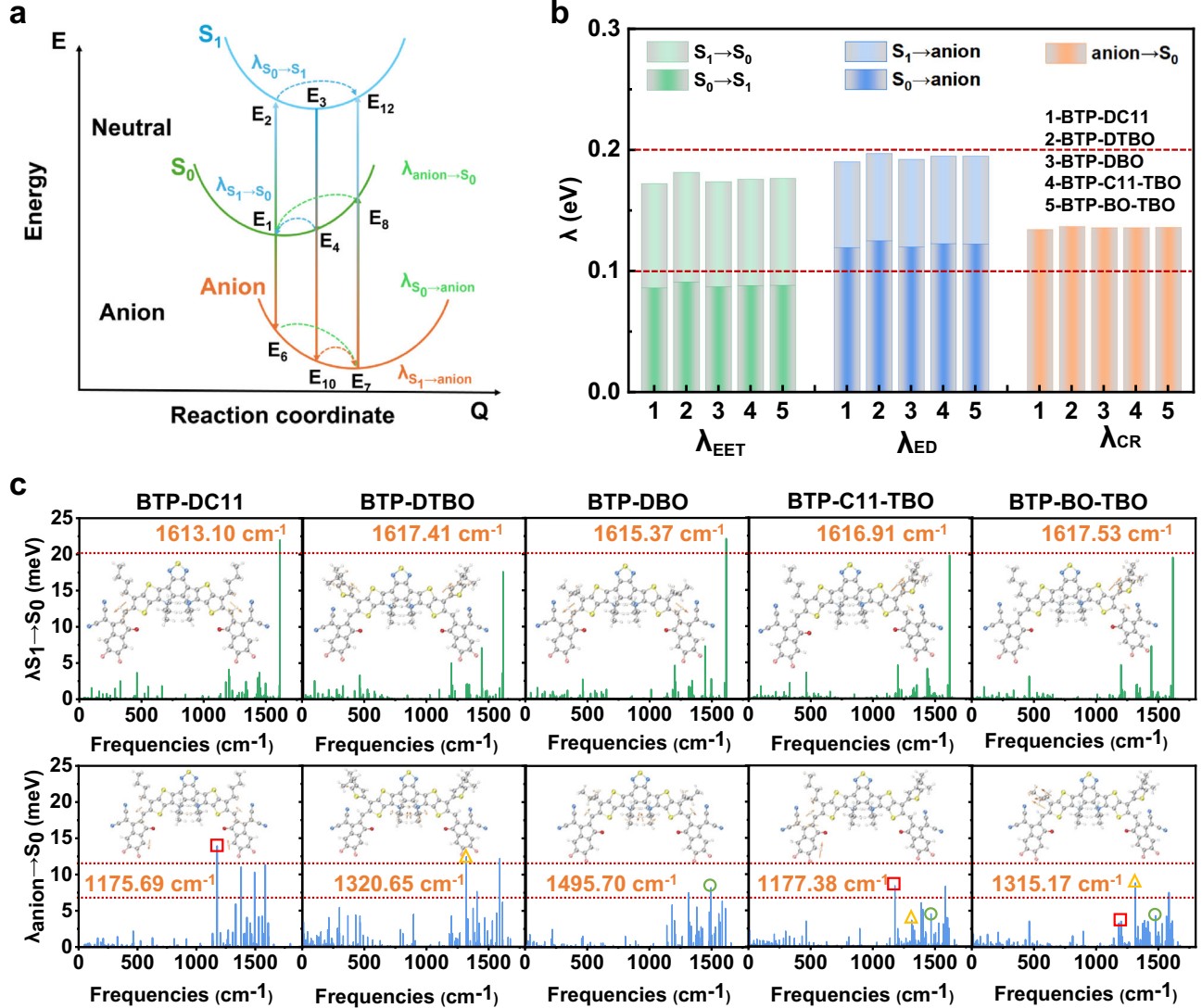

**Fig. 2 | Reorganization energy of acceptors. a** Schematic diagram of the related transitions among the $S_0$, $S_1$, and the anionic state during the photoelectric conversion processes. **b** The corresponding reorganization energies of BTP-DC11, BTP-DTBO, BTP-DBO, BTP-C11-TBO, and BTP-BO-TBO acceptors at the level of ωB97XD/ 6-31G (d, p). **c** Contributions of each vibrational mode to the reorganization energy for the $S_1{\rightarrow}S_0$ and anion→$S_0$ transitions of five SMAs, and the displacement vectors for the vibrational normal modes marked by squares (at around 1170 cm⁻¹), triangle (at around 1320 cm⁻¹) and circles (1495 cm⁻¹) are inserted. The height of displacement vectors stands for the magnitude of vibrational strength.

decreased, suggesting the fast exciton separation and diffusion, which is conducive to obtaining a higher short-circuit current density ($J_{SC}$). Interestingly, BTP-BO-TBO-based blend increases slowly to the maximum value and then decays slowly, demonstrating the longer exciton diffusion lifetime and CT state lifetime, which also indicates a significant reduction of charge recombination in the blend (Fig. 3c). In other words, the charge separation and recombination are limited by exciton diffusion, benefiting in the higher FF for PM6:BTP-BO-TBO-based devices. Furthermore, to get an in-depth understanding of the exciton lifetimes of these acceptor materials, transient fluorescence measurements were tested to obtain the exciton lifetime of the pristine films. By fitting the attenuation curve with a double exponential (as shown in Supplementary Fig. 10 and Supplementary Table 3 in the Supplementary Information), it can be found that all the acceptors showed similar $t_1$ (50 ps) and $t_2$ (300 ps) values, suggesting the similar exciton lifetime of these thin acceptor films. Besides, considering that the exciton lifetime is also closely related to the molecular packing features, we suspect that the BTP-BO-TBO component in the PM6:BTP-BO-TBO blend possesses a

longer exciton diffusion length, thus resulting in a longer exciton diffusion lifetime.

According to the classical Marcus electron transfer model[36], the exciton dissociation rate constants ($\tau_1$) follow the formula:

$$\frac{1}{\tau_1} = \frac{2\pi}{\hbar\sqrt{4\pi\lambda k_B T}} V^2 \exp\left(-\frac{(\lambda + \Delta G)^2}{4\lambda k_B T}\right) \quad (2)$$

where $1/\tau_1$ refers to the hole-transfer rate; $k_B$ is the Boltzmann constant; $T$ is the temperature; $V$ is the electronic coupling between initial and final states; $\Delta G$ is the change in free energy. By fitting the hole-transfer kinetics with a bi-exponential function, the BTP-DC11, BTP-DTBO, BTP-DBO, BTP-C11-TBO, and BTP-BO-TBO-based blends exhibited fast $\tau_1$ values of 0.51, 0.50, 0.48, 0.57, and 0.49 ps, respectively, and slow $\tau_2$ values of 2.5, 15, 5.2, 3.8, and 12 ps, respectively (Supplementary Fig. 11 and Supplementary Table 4). Typically, the slower $\tau_2$ is thought to represent the time required for exciton diffusion toward the interface prior to dissociation, while the ultrafast $\tau_1$ is thought to represent the

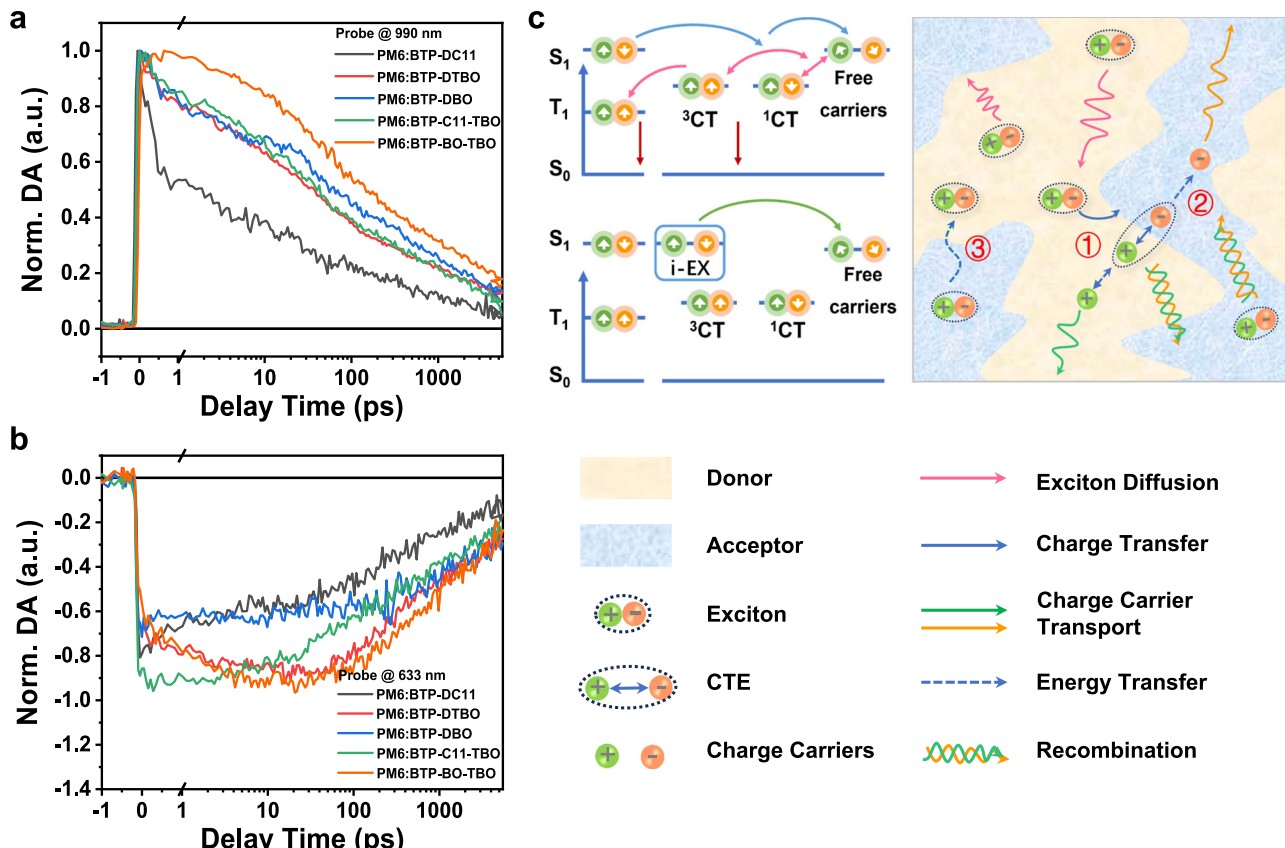

**Fig. 3 | Excited-state kinetics revealed by transient absorption spectroscopy.** **a** Kinetic traces probing at 633 nm for the GSB and **b** TA traces of the five blends probed at 990 nm of PM6:BTP-DC11, PM6:BTP-DTBO, PM6:BTP-DBO, and PM6:BTP-C11-TBO and PM6:BTP-BO-TBO blend films. **c** Schematic of the behaviors of exciton and charge carriers.

time needed for acceptor exciton dissociation at the D/A interface. Results reveal that the introduction of the thienyl outer side chains marginally decreased the exciton transfer rate at the D:A interface, whereas it significantly prolonged a portion of the exciton diffusion-mediated transfer process, which may be related to the existence of larger pure acceptor domains and a smaller mixed-phase[37]. Therefore, to evaluate the differences in domain purity among these blend films, the surface energies (SEs) and corresponding Flory-Huggins interaction parameter ($\chi$) values of these materials were calculated by contact angle measurements, as shown in Supplementary Fig. 12 and Supplementary Table 5. The closer the SE means the better the miscibility between the donor and acceptor[38]. According to the empirical formula:

$$\chi = K(\sqrt{\gamma_D} - \sqrt{\gamma_A})^2 \tag{3}$$

where K is a constant, $\gamma_D/\gamma_A$ represents the SE of the donor/acceptor, and the $\chi$ values of the donor PM6 and five SMAs blends were calculated to be 0.28 K, 0.64 K, 0.49 K, 0.62 K, and 0.74 K, respectively. Among them, the $\chi$ of PM6: BTP-BO-TBO blend film presented the highest value, indicating the higher domain purity of the BTP-BO-TBO-based blend film. As mentioned above, the asymmetric molecular geometry can promote exciton diffusion charge transfer while inhibiting charge recombination, thereby reducing the non-radiative voltage loss[39,40].

Recent studies demonstrated that enhancing the hybridization of the LE and CT states is conducive to suppressing the non-radiative recombination, thereby reducing the non-radiative voltage loss ($\Delta V_{nr}$) of the OSC devices[41]. As the strength of the hybridization of LE and CT states is closely related to the energy offset between LE and CT states ($\Delta E_{LE-CT}$). The smaller $\Delta E_{LE-CT}$ indicates the stronger

hybridization of LE and CT states. Therefore, a combination of theoretical and experimental approaches was applied to accurately evaluate the effects of asymmetric molecular structures on the hybridization of LE and CT states. Generally, the low-energy shift of the blended system reduces the driving force for exciton dissociation to charge-transfer (CT) states, which is unfavorable for the effective dissociation of exciton[17]. Thus, the driving force ($\Delta E_{LE-CT}$) for exciton dissociation can be considered as the energy difference between the CT states and locally excited (LE) states (Fig. 4a)[42,43]. Gaussian fitting for the highly sensitive EQE and EL in the charge-transfer absorption region was performed to acquire $E_{CT}$ according to the Marcus equations, as follows[44]:

$$EQE_{PV}(E) = \frac{f_j}{E\sqrt{4\pi\lambda_j kT}} \exp\left(-\frac{(E_{CT} + \lambda_j - E)^2}{4\lambda_j kT}\right) \tag{4}$$

$$EQE_{EL}(E) = \frac{Ef_j}{\sqrt{4\pi\lambda_j kT}} \exp\left(-\frac{(E_{CT} - \lambda_j - E)^2}{4\lambda_j kT}\right) \tag{5}$$

where $f_j$ refers to the transfer integral of electronic coupling between the CT and LE states, and $\lambda_j$ represents the reorganization energy. As a result, the $E_{CT}$ values were 1.42, 1.38, and 1.38 eV for BTP-DBO-, BTP-DTBO- and BTP-BO-TBO-based devices, respectively[45], and calculated $\Delta E_{LE-CT}$ values of the PM6:BTP-BO-TBO-based was slightly smaller than PM6:BTP-DBO and PM6:BTP-DTBO-based devices (Fig. 4b-d). Depressed $\Delta E_{LE-CT}$ contributes to exciton transfer from the CT state back to the LE state, improves the luminescence efficiency, and reduces the non-radiative recombination[46].

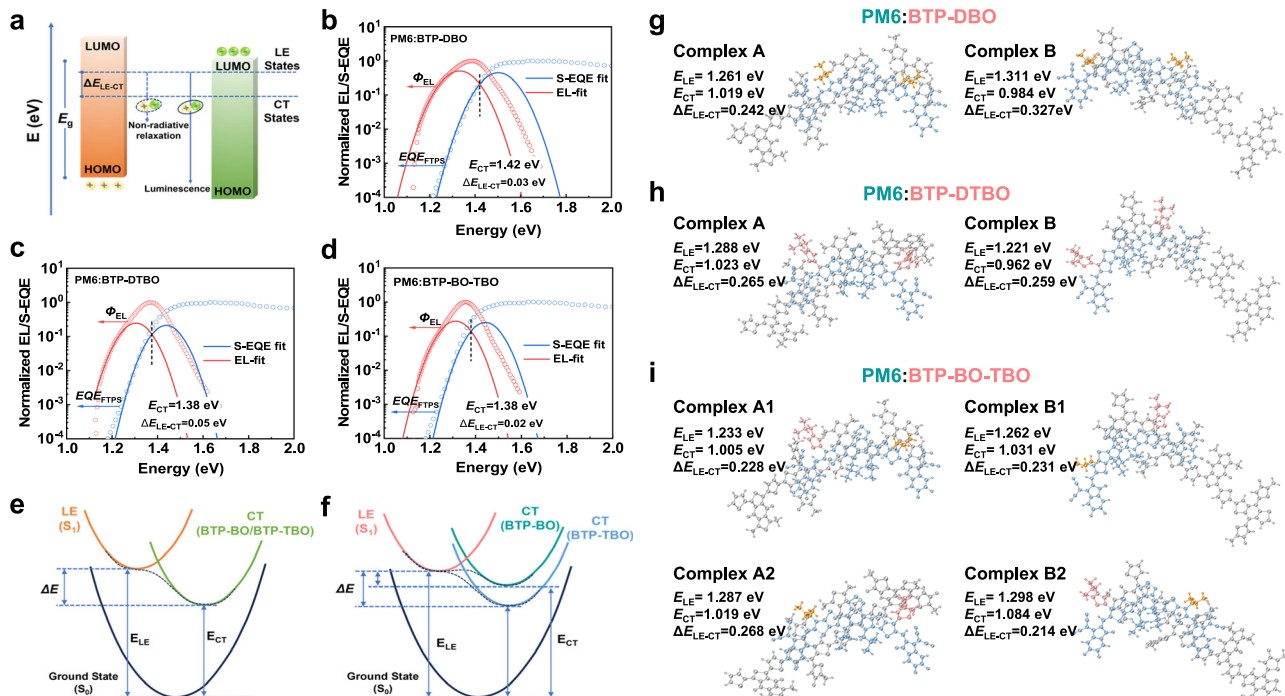

**Fig. 4 | The calculated state energies of D: A complex configuration. a** Energy level diagram of CT and LE states, and correlation $\Delta E_{LE\text{-}CT}$ of the binary system. **b–d** Gaussian fits of sEQE and EL curves for devices based on PM6:BTP-DBO, PM6:BTP-DTBO, and PM6:BTP-BO-TBO. **e, f** Diagrams of energy differences. **g–i** The complexes of PM6:BTP-DBO, PM6:BTP-DTBO, and PM6:BTP-BO-TBO and the related $E_{LE}$, $E_{CT}$, and $\Delta E_{LE\text{-}CT}$.

To further gain a clear understanding of the effects of the acceptor side chain structures on the nature of their CT states, TD-DFT calculations on the D: A complexes of two symmetric acceptor-based blends (PM6:BTP-DBO and PM6:BTP-DTBO) and one asymmetric acceptor-based blend (PM6:BTP-BO-TBO) were performed to characterize the nature of their excited states, as shown in Fig. 4e, f, the source data are provided in Supplementary Data 6–13. For the two symmetric acceptors (BTP-DBO and BTP-DTBO), two types of complexes could be obtained depending on whether the side chain group is close to the BDT or BDD unit of PM6. Besides, it could be obtained that, the conformations of BTP-DBO-based complexes differed from those of BTP-DTBO-based complexes, suggesting the significant effects of side chain structures of acceptors on the D: A intermolecular interactions. In the case of the asymmetric acceptor BTP-BO-TBO, four different types of complexes were generated depending on whether the alkyl side chain or thiophene side chain of the acceptor is close to the BDT or BDD moiety of PM6.

As seen from Fig. 4g–i, despite the symmetric molecular structures, the two complexes (complex A and complex B) based on BTP-DBO or BTP-DTBO present different LE and CT state energies. As a consequence, both BTP-DBO and BTP-DTBO-based complexes exhibit two different $\Delta E_{LE\text{-}CT}$ values (0.242 eV and 0.327 eV for PM6:BTP-DBO, 0.265 eV and 0.259 eV for PM6:BTP-DTBO). While for the PM6:BTP-BO-TBO-based complexes (complex A1 and complex A2) with the side chain close to the BDT unit, the $\Delta E_{LE\text{-}CT}$ values are similar to the corresponding complexes of BTP-DBO (complex A) and BTP-DTBO (complex A), with a smaller $\Delta E_{LE\text{-}CT}$ of 0.228 eV. Similarly, for the complexes (complex B1 and complex B2) with the side chain close to the BDD unit, a smaller $\Delta E_{LE\text{-}CT}$ value of 0.214 eV could also be obtained, in comparison with the corresponding parts of BTP-DBO (complex B) and BTP-DTBO (complex B).

The $\Delta E_{LE\text{-}CT}$ value is recognized as a factor in determining the degree of LE-CT electronic hybridization and the magnitude of the charge recombination processes[46]. Calculations indicate that the $\Delta E_{LE\text{-}CT}$ values of the PM6:BTP-BO-TBO-based complexes are smaller than those

of symmetric acceptors and expected to have a stronger hybridization between the CT and LE states, which contributes to the observation of smaller $\Delta E_{nr}$ in PM6:BTP-BO-TBO-based devices[17]. In addition, it should be noted that, despite the slightly smaller $\Delta E_{LE\text{-}CT}$ values, the asymmetric acceptor-based complexes also possess similar $\Delta E_{LE\text{-}CT}$ values with the symmetric acceptor-based, indicating the sufficient driving force for exciton dissociation. In brief, these features highlight that the dual nature of the interfacial structural and electronic properties, thanks to the asymmetric design of BTP-BO-TBO, may be the basis for rapid exciton dissociation and small non-radiative voltage loss in blends[36,47].

Considering the energy migration between the donor and acceptor molecular states coupled with the hybridization of the CT exciton formed after the initial charge transfer with the emitted LE state, results in slow charge separation, as observed in transient absorption spectra, and also minimizes non-radiative relaxation to the ground state. Therefore, it is helpful in the above study low $\Delta E_{LE\text{-}CT}$ OSC systems show small non-radiative recombination and high $V_{OC}$. Next, we will investigate a series of device characterizations.

## Photovoltaic characteristics and energy loss

To investigate the effect of asymmetric side chains on the photovoltaic performance of OSCs, we fabricated the conventional binary device structure with the ITO /PEDOT: PSS /PM6:SMAs /PDINN/Ag, where polymer PM6 was selected as the donor due to its complementary absorption and energy level matching with five acceptors. Details of device preparation are shown in the "Method" section, and the optimal $J$–$V$ curve and external quantum efficiency (EQEs) spectrum are shown in Fig. 5a, b, and Supplementary Tables 6 and 7.

As summarized in Table 2, the incorporation of the branched alkyl side chain and thiophene-based side chain led to the gradual increase in the $V_{OC}$ values, that is, the $V_{OC}$ of PM6:BTP-BO-TBO (0.913 V) is significantly higher than that of PM6:BTP-C11-TBO (0.856 V) and PM6:BTP-DTBO (0.881 V). Notably, the devices based on asymmetric acceptors BTP-BO-TBO and BTP-C11-TBO realize satisfactory PCEs of 19.76% and 18.51%, respectively, greater than those devices based on

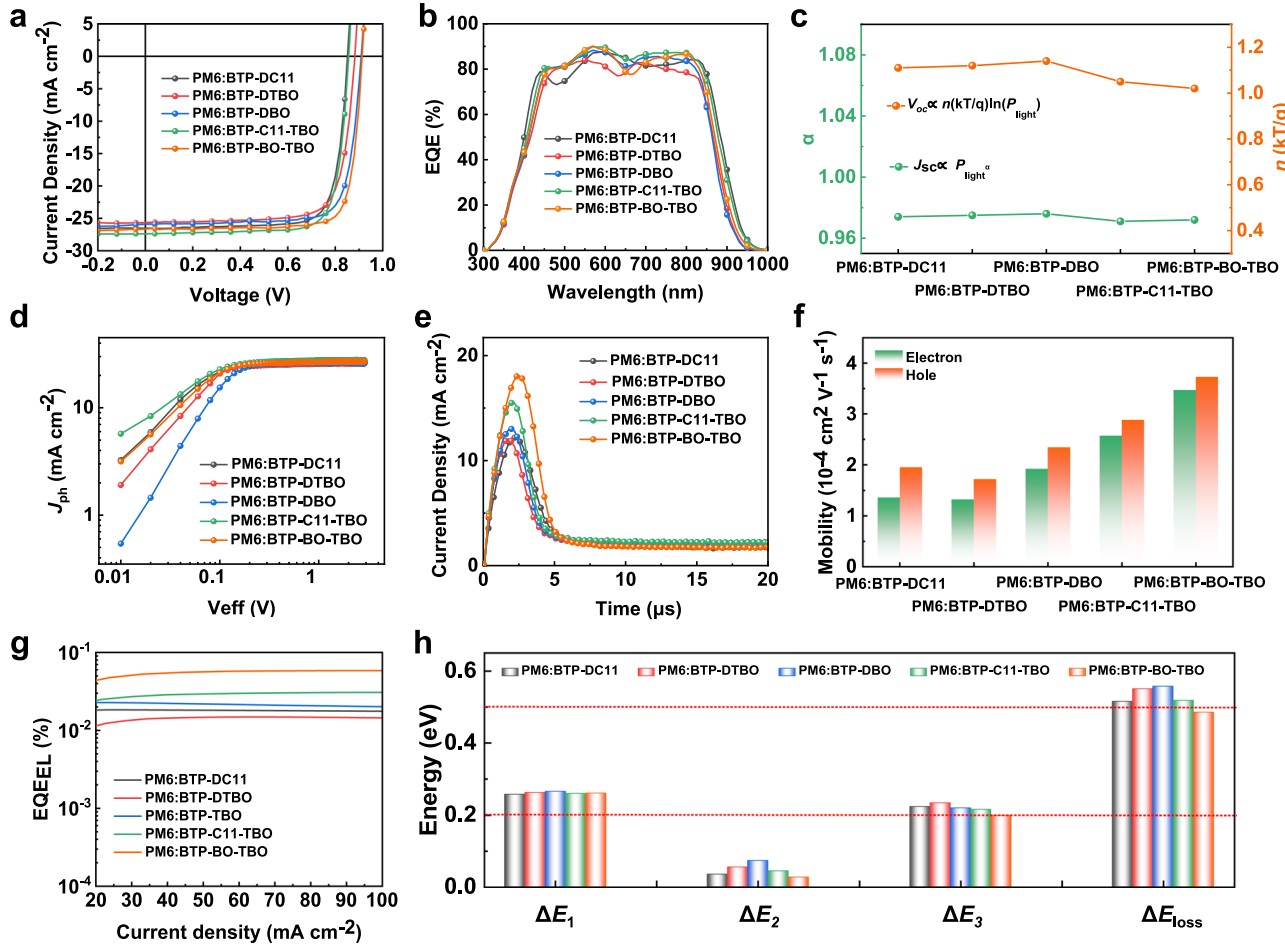

**Fig. 5 | Photovoltaic properties of OSCs. a** $J–V$ curves of the optimal OSCs. **b** EQE curves of the corresponding devices. **c** Dependences of $V_{OC}$ and $J_{SC}$ on $P_{light}$. **d** $J_{ph}$ versus $V_{eff}$ curves of the OSCs. **e** Photo-CELIV characteristics of OSCs **f** Hole and electron mobilities of the optimized devices. **g** $EQE_{EL}$ curves of the optimal OSCs **h** Statistical diagram of $E_{loss}$ of OSCs.

**Table 2 | Photovoltaic performance parameters of the optimized OSCs under the illumination of AM 1.5 G, 100 mW cm⁻²**

| Device | $V_{OC}$ [V] | $J_{SC}$ [mA cm⁻²] | $J_{SC}^a$ [mA cm⁻²] | FF [%] | PCE$^b$ [%] |
|---|---|---|---|---|---|
| PM6:BTP-DC11 | 0.852 | 26.50 | 26.12 | 78.69 | 17.76 (17.45 ± 0.15) |
| PM6:BTP-DTBO | 0.881 | 25.64 | 24.93 | 76.23 | 17.22 (16.88 ± 0.27) |
| PM6:BTP-DBO | 0.909 | 25.89 | 25.16 | 78.27 | 18.42 (18.11 ± 0.25) |
| PM6:BTP-C11-TBO | 0.856 | 27.35 | 26.71 | 79.06 | 18.51 (18.24 ± 0.13) |
| PM6:BTP-BO-TBO | 0.913 | 26.67 | 25.89 | 81.17 | 19.76 (19.39 ± 0.21) |

ᵃCalculated from EQE curves.

ᵇThe values in parentheses are the average values with standard deviations obtained from 20 devices.

symmetric acceptors, particularly the OSC of PM6:BTP-BO-TBO exhibits a splendid FF of 81.17% and a decent $J_{SC}$ of 26.67 mA cm⁻². The enhanced $J_{SC}$ and FF values in the BTP-BO-TBO-based device might arise from the favorable molecular stacking and better charge transport capability of the PM6:BTP-BO-TBO blend active layer. To our knowledge, the impressive PCE of 19.76% and FF of 81.17% are among the highest values for binary OSCs based on asymmetric acceptors. As depicted in Fig. 5b, almost all the five optimal devices exhibit high EQE values (exceeding 80%), suggesting efficient internal carrier conversion (or charge transfer/collection). It is important to note that the PM6:BTP-C11-TBO-based device has a broader EQE response relative to the PM6:BO-TBO-based device, obtaining an integrated $J_{SC}$ of 26.71 mA cm⁻², which should be expected to derive from the narrower bandgap of BTP-C11-TBO. The EQE profile of the PM6:BTP-BO-TBO-

based device exhibits a slight blue shift versus the device based on PM6:BTP-C11-TBO, leading to a slightly lower $J_{SC}$ in the devices. The operational stability of the two asymmetric acceptor molecules (PM6:BTP-C11-TBO and PM6:BTP-BO-TBO) based OSCs have been measured. Both of them retained above 95% initial efficiency after 600 s maximum power point tracking under continuous AM 1.5 G illumination. In addition, after storage in a nitrogen-filled glovebox for 1600 h, both BTP-C11-TBO and PM6:BTP-BO-TBO-based OSCs devices retained more than 80% of their initial PCEs. Furthermore, upon continuous white LED irradiation for 250 h, both of the devices maintained more than 90% of their initial PCEs. These results demonstrate the preferable device stability of BTP-C11-TBO and PM6:BTP-BO-TBO-based OSCs (as shown in Supplementary Figs. 13 and 14 in the Supplementary Information).

**Table 3 | Total energy losses and detailed energy losses of the optimized devices**

| Active layer | $E_g$ [eV] | $V_{OC}$ [V] | $E_{loss}$[a] [eV] | $\Delta E_1$ [eV] | $\Delta E_2$ [eV] | $\Delta E_3$ [eV] | $EQE_{EL}$ [×10$^{-4}$] |
|---|---|---|---|---|---|---|---|
| PM6:BTP-DC11 | 1.366 | 0.851 | 0.515 | 0.257 | 0.035 | 0.223 | 1.84 |
| PM6:BTP-DTBO | 1.428 | 0.878 | 0.550 | 0.262 | 0.055 | 0.233 | 1.22 |
| PM6:BTP-DBO | 1.457 | 0.900 | 0.557 | 0.265 | 0.073 | 0.219 | 2.27 |
| PM6:BTP-C11-TBO | 1.374 | 0.856 | 0.518 | 0.259 | 0.044 | 0.215 | 2.51 |
| PM6:BTP-BO-TBO | 1.392 | 0.907 | 0.485 | 0.260 | 0.027 | 0.198 | 4.75 |

[a]The calculation processes of $E_{loss}$ are shown in the "Methods" section and Supplementary Figs. 17–18.

The relationship between light intensity ($P_{light}$) and $V_{OC}$ as well as $J_{SC}$ was investigated to elucidate the dynamics of charge recombination. Focusing on the relationship of $V_{OC}$ *vs* ln $P_{light}$, the slopes of the devices BTP-DC11-, BTP-DTBO-, BTP-DBO-, and BTP-C11-TBO- and BTP-BO-TBO-based are calculated to be 1.11, 1.12, 1.14, 1.07 and 1.02 $kT/q$, respectively (Fig. 5c and Supplementary Fig. 15a), demonstrating the minimal trap-assisted recombination for PM6:BTP-BO-TBO device[26]. Figure 5c and Supplementary Fig. 15b present the dependence of $J_{SC}$ and $P_{light}$ following the relation of $J_{SC} \propto P_{light}{}^a$, where the slope ($a$) of all devices is close to 1, meaning that bimolecular recombination losses are negligible for the five devices[48,49]. Meanwhile, the exciton dissociation probability ($P_{diss}$) can be obtained by investigating the relation between the effective voltage ($V_{eff}$) and the saturated photocurrent density ($J_{ph}$) in the five devices (Fig. 5d). The $P_{diss}$ values of five devices are 85.1% of PM6:BTP-DC11, 87.1% of PM6:BTP-DTBO, 90.2% of PM6:BTP-DBO, 93.1% of PM6:BTP-C11-TBO and 95.8% of PM6:BTP-BO-TBO, respectively. The PM6:BTP-BO-TBO-based OSCs manifest the optimal exciton dissociation ability, which could be responsible for its higher FF.

To shed more light on the asymmetric side chain influence on charge mobility, we looked into the electron mobility ($\mu_e$) and the hole mobility ($\mu_h$) of the blend active layers in detail by the space-charge limited current (SCLC) measurements. As exhibited in Supplementary Table 8, the highest $\mu_e$ of $3.47 \times 10^{-4}$ cm$^2$ V$^{-1}$ s$^{-1}$ and $\mu_h$ of $3.73 \times 10^{-4}$ cm$^2$ V$^{-1}$ s$^{-1}$ are achieved for the PM6:BTP-BO-TBO blend due to the significantly enhanced molecular packing (Fig. 5f and Supplementary Fig. 16). Furthermore, the $\mu_e/\mu_h$ ratios of devices for BTP-DC11-, BTP-DTBO-, BTP-DBO-, BTP-C11-TBO and BTP-BO-TBO-based are 0.70, 0.77 0.83, 0.89, and 0.93, respectively. Moreover, we further evaluated the charge mobility via photo-CELIV measurements (Fig. 5e), where the PM6:BTP-BO-TBO-based device presents the highest charge mobilities, consistent with the results of the SCLC method. The higher charge mobilities and more balanced $\mu_e/\mu_h$ in the PM6:BTP-BO-TBO device are responsible for the better FF of its OSCs.

To investigate the cause of $V_{OC}$ enhancement in devices based on asymmetric acceptor molecules, we carried out $E_{loss}$ analysis by utilizing highly sensitive EQE (sEQE) and electroluminescence (EL). The overall $E_{loss}$ of OSCs resulted from three aspects: the charge recombination generated by unavoidable black body radiation ($\Delta E_1$), radiative recombination loss from below-gap absorption ($\Delta E_2$), and non-radiative recombination loss ($\Delta E_3$)[50,51]. Detailed $E_{loss}$ analyses are presented in Supplementary Fig. 17 and Table 3. The charge-transfer state energy ($E_{CT}$) of devices based on BTP-DC11, BTP-DTBO, BTP-DBO, BTP-C11-TBO, and BTP-BO-TBO were 1.34, 1.38, 1.42, 1.36, and 1.38 eV, respectively, accounting for slightly different energetic differences between the singlet excited states and the charge-transfer states ($\Delta E_{CT}$) (Supplementary Fig. 18 and Supplementary Table 9)[52,53]. Moreover, the electroluminescence quantum efficiencies ($EQE_{EL}$) of the devices are shown in Fig. 5g, and the $EQE_{EL}$ values of the five devices are measured to be $1.84 \times 10^{-4}$ for PM6:BTP-DC11, $1.22 \times 10^{-4}$ for PM6:BTP-DTBO, $2.27 \times 10^{-4}$ for PM6:BTP-DBO, $2.51 \times 10^{-4}$ for PM6:BTP-C11-TBO, and $4.75 \times 10^{-4}$ for PM6:BTP-BO-TBO, respectively, among which the $EQE_{EL}$ of the PM6:BTP-BO-TBO device is higher than those of the other devices.

According to the equation:

$$\Delta E_{nr} = -\frac{KT}{q}\ln(EQE_{EL}) \qquad (6)$$

The $\Delta E_{nr}$ of these devices could be obtained, as presented in Table 3. Ultimately, the devices constructed by PM6: BTP-BO-TBO achieved the lowest $\Delta E_{nr}$ of 0.198 eV and the smaller $E_{loss}$ values of 0.485 eV (see Fig. 5h). Subsequently, we calculated the Urbach energy ($E_U$) by fitting the sEQE spectra beyond the bandgap edge with a linear equation (see Supplementary Fig. 19)[54]. The $E_U$ values of BTP-C11-TBO- and BTP-BO-TBO-based devices were 23.27 and 22.39 meV, respectively, which were lower than those of BTP-DC11-, BTP-DTBO-, and BTP-DBO-based devices (23.43, 24.37, and 25.26 meV, respectively), revealing the PM6:BTP-BO-TBO blend film has a lower energetic disorder. Overall, the alkyl/thienyl asymmetric side chain is an effective strategy to reduce the total $E_{loss}$ and increase the $V_{OC}$ values without sacrificing photocurrent.

## Film morphology

The microscopic morphology of the acceptor materials and the active layer films of the OSCs play a crucial role in their photoelectric conversion efficiency. To acquire a profound understanding of the influence of various side chains on the film morphology of the active layers, atomic force microscopy (AFM), transmission electron microscopy (TEM), and scanning transmission electron microscopy (STEM) were conducted on these PM6:SMAs blend films. As depicted in Fig. 6a, compared with the blends based on BTP-DC11 and BTP-DBO, the root-mean-square roughness (RMS) of the blends based on BTP-C11-TBO (0.903 nm) and BTP-BO-TBO (0.868 nm) decreased after the introduction of thiophene side chains, especially that of PM6:BTP-BO-TBO, showing a more uniform and smoother surface characteristics. Moreover, the TEM images of PM6:BTP-BO-TBO showed a more dense and uniform black-and-white distribution compared with other blend films (Fig. 6b and Supplementary Fig. 20). It is worth noting too that the STEM images of PM6:BTP-BO-TBO exhibited a clearer and more pronounced phase separation structure characteristics of the fibrous 3D network showed in Supplementary Fig. 21. Additionally, it is interesting to note that the STEM images of PM6:C11-TBO and PM6:BO-TBO are relatively consistent with those of PM6:BTP-DC11 and PM6:BTP-DBO, respectively, that is that PM6:C11-TBO and PM6:BTP-DC11 show cluster-like aggregation, while PM6:BO-TBO and PM6:DBO mainly exhibit fibrous-like aggregation, indicating that the morphology regulation of donor–acceptor blends based on side-chain asymmetric acceptor is mainly affected by aliphatic alkyl chains while the thiophene alkyl chain had little effect. The above results manifest that modifying the side chains of SMAs is conducive to achieving the preferable phase separation morphology of blend films, promoting charge transport and improving FF, thereby optimizing photovoltaic performance.

To fully understand the difference in the stacking behavior of SMAs with side-chain symmetric and asymmetric, we first explored the single-crystal structure[55,56] (see Supplementary Figs. 22–24 and

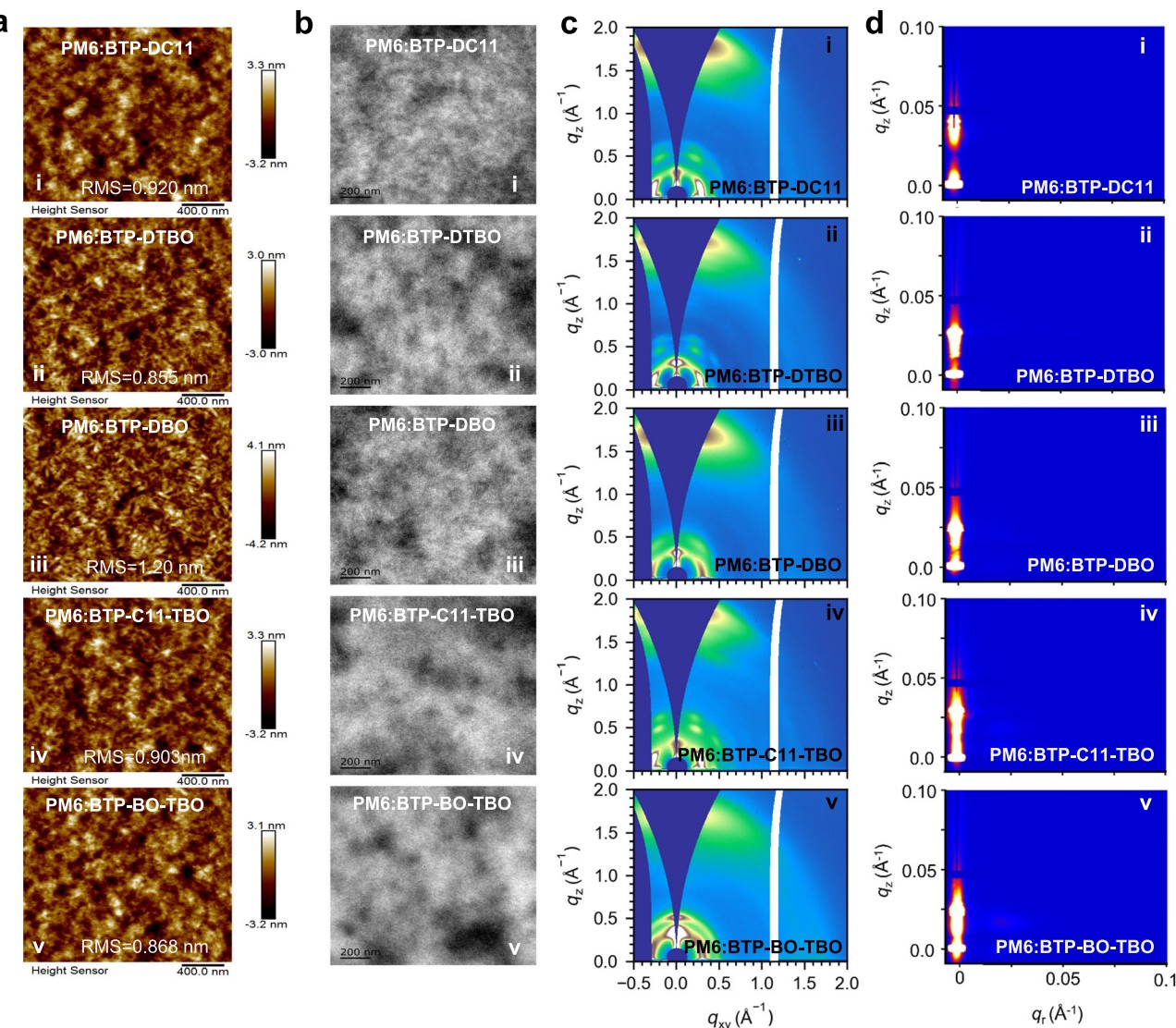

**Fig. 6 | Morphology features of the blend films. a** AFM height images of the optimal blend films. **b** TEM images of the optimal blend films. **c** 2D GIWAXS patterns and **d** 2D GISAXS patterns of the corresponding films. (i:PM6:BTP-DC11; ii:PM6:BTP-BTP-DTBO; iii:PM6:BTP-DBO; iv:PM6:BTP-C11-TBO; v:PM6:BTP-BO-TBO).

Supplementary Table 10), the crystallographic data are provided in Supplementary Data 14–16. Compared with BTP-DBO, BTP-DTBO and BTP-BO-TBO both have four different stacking modes in a single unit cell, providing a richer transport channel for charge transfer. Meanwhile, the single-crystal structure showed that tMode 2: A/A and Mode 3:D-A/A-D in BTP-BO-TBO have a shorter π–π distance than BTP-DTBO (Supplementary Fig. 25), which indicates that the introduction of asymmetric side chains can promote formation tighter π–π packing. Subsequently, we calculated electronic coupling ($|J|$) based on a single-crystal structure to estimate the charge transfer among SMAs[57]. As shown in Supplementary Fig. 26, taking 1 as a reference molecule, the adjacent molecules named 2, 3, 4, etc., respectively, the electron-coupled pairs formed are represented by 1–2, 1–3, 1–4…, etc. The BTP-DBO, BTP-DTBO, and BTP-BO-TBO single crystals all possess 4 nearest neighbors and 4 electron-coupled pairs, and the detailed $|J|$ are summarized in Supplementary Table 11. The relative order of the overall $|J|$ values of the three molecules is as follows: $|J|_{(BTP-DTBO)} > |J|_{(BTP-BO-TBO)} \geq |J|_{(BTP-DBO)}$. Significantly, however, the $|J|_{max}$ of BTP-BO-TBO is located 1–2 dimer (Mode 1:D-A/A-D), rather than the 1–4 dimer (Mode 3:D-A/A-D) in BTP-DTBO and BTP-DBO single crystals. The former has a larger stacking area between molecules, which is more conducive to charge transfer[58].

The grazing-incidence wide-angle X-ray scattering (GIWAXS) was utilized to probe the molecular packing and crystallinity behaviors. As exhibited in Supplementary Figs. 27 and 28 and Supplementary Table 12, the five SMAs neat films all exhibit obvious (100) lamellar stacking peaks in the in-plane (IP) direction and ordered π–π stacking (010) peaks in the out-of-plane (OOP) direction, and manifest a significant face-on stacking orientation. The diffraction signals of the alkyl/thienyl asymmetric side-chain acceptors BTP-C11-TBO and BTP-BO-TBO were more similar to those of the symmetric alkyl side-chains acceptors BTP-DC11 and BTP-DBO, respectively, indicating that the thienyl side-chain diffraction signals of the asymmetric SMAs BTP-C11-TBO and BTP-BO-TBO were not prominent. A more interesting thing is that the one-dimensional diffraction patterns of BTP-DBO and BTP-BO-TBO acceptors molecules show two connected scattering in the (010) peaks in the OOP direction, indicating that the strong crystallinity of their molecules leads to periodic diffraction peaks in the direction of π–π accumulation[59]. Nevertheless, the asymmetric acceptor BTP-BO-TBO exhibits a smaller π–π packing *d*-spacing than the symmetric molecules (BTP-DBO and BTP-DTBO), (BTP-DC11: *d*-spacing = 3.713 Å, CCL = 12.68 Å; BTP-DTBO: *d*-spacing = 3.891 Å, CCL = 15.56 Å; BTP-DBO: *d*-spacing = 3.556 Å, CCL = 29.67 Å; BTP-C11-TBO: *d*-spacing = 3.758 Å, CCL = 14.74 Å; BTP-BO-TBO: *d*-spacing = 3.534 Å, CCL = 28.21 Å),

indicating that the introduction of the alkyl/thienyl asymmetric side chains could lead to more tightly ordered molecular packing of the BTP-BO-TBO.

Supplementary Figs. 27 and 28 show the 2D diffraction and corresponding 1D linecuts pattern of PM6 neat film, respectively. There is a strong (100) scattering peak ($q_r$ = 0.322 Å$^{-1}$) along the IP direction while (010) relatively weak π–π stacking diffraction peak ($q_z$ = 1.639 Å$^{-1}$) in the OOP direction, the whole showing a weak face-on packing orientation. Blended PM6 with these five SMAs, the face-on molecular orientation of all the blends gain remarkably strengthened with (100) peak $q_r$ at ca. 0.33 Å$^{-1}$ and (010) peak $q_z$ at ca. 1.66 Å$^{-1}$ (see Fig. 6c and Supplementary Table 13). Significantly, PM6:BTP-C11-TBO and PM6:BTP-BO-TBO exhibit similar diffraction signals to PM6:BTP-DC11 and PM6:BTP-DBO, respectively, which aligns closely with the results of neat SMAs, suggesting that the alkyl/thienyl asymmetric side chain substituted SMAs possess the similar molecular properties (including molecular stacking behavior and/or miscibility), with their corresponding alkyl-substituted symmetric SMAs. In addition, the blend film of PM6:BTP-BO-TBO was more tightly packed versus PM6:BTP-C11-TBO in the (010) region (BTP-C11-TBO: d-spacing = 3.820 Å, CCL = 18.08 Å; BTP-BO-TBO: d-spacing = 3.760 Å, CCL = 18.87 Å), indicating the maintainable chain-extended feature and comparatively large crystalline domain of BTP-BO-TBO, which facilitates charge transport within the molecular backbone, thus increasing electron mobility. Meanwhile, in order to in-depth inquiring the reasons for BTP-BO-TBO-based device enhanced FF, we also conducted the small-angle X-ray scattering (GISAXS) to study the phase segregation of the five blends. Figure 6d presents the 2D patterns and Supplementary Fig. 29 displays the IP direction intensity profiles fitted by a fractal-like network model and the Debye–Anderson–Brumberger (DAB) model is employed for quantization of the size of the amorphous intermixing region[60]. Although the average domain sizes (2R$_g$) of the BTP-C11-TBO and BTP-BO-TBO-based blends are within the proper range for effective exciton dissociation (<30 nm), the larger intermixing domain size (26 nm) of BTP-BO-TBO-based films is conducive to gaining higher domain purity, thereby improving charge transport, inhibiting charge recombination, and obtaining higher FF values[61] (the data are summarized in Supplementary Table 14).

## Discussion

In summary, we have demonstrated that the alkyl/thienyl asymmetric side chain strategy offers an effective method to suppress exciton-vibration coupling and reduce the offset between the CT and LE states, thus realizing inhibited non-radiative loss without sacrificing the charge collection efficiency. The two asymmetric SMAs (BTP-C11-TBO and BTP-BO-TBO) with alkyl/thienyl asymmetric outer side chains demonstrated fine-tuned molecular packing and blend morphology. With the introduction of the thienyl side chain, BTP-C11-TBO and BTP-BO-TBO neat thin films exhibit slightly larger optical bandgaps, elevated LUMO energy levels, and facilitated electron mobility. Liken to BTP-DC11 with a straight alkyl chain, the alkyl/thienyl asymmetric SMAs exhibit enhanced crystallization, contributing to the tighter π–π packing distances along the conjugated backbone. As a consequence, benefiting from the specific electron-vibration coupling properties, packing characteristics, and electronic structure features caused by the alkyl/thienyl asymmetric side chains, the OSCs based on PM6:BTP-BO-TBO delivered the lowest non-radiative energy loss of 0.198 eV with efficient exciton dissociation and charge collection efficiency, thus contributing to its high $V_{OC}$ and $J_{SC}$, as well as the champion PCE of 19.76%, which is one of the highest values for binary OSCs. This work highlights the role of alkyl/thienyl asymmetric side chain strategy in reducing the non-radiative recombination energy, realizing efficient charge generation, suppressing charge recombination, and thus improving the photovoltaic performance of OSCs.

## Methods

### Materials

The detailed synthesis process of BTP-C11-TBO and BTP-BO-TBO and the corresponding structural characteristics were described in the Supplementary information. Other materials are provided by commercial suppliers: PM6 (Solarmer Energy Inc.), Pd (PPh$_3$)$_4$, and 1-chloronaphthalene (J& K Chemical Co.), and these reagents and solvents purchased are used directly without further purification.

### Molecular structure characterization

$^1$H NMR and $^{13}$C NMR spectra were recorded using Bruker AVANCE NEO 600 MHz spectrometer in d-chloroform solution. Chemical shifts were reported as δ values (ppm) with tetramethyl silane (TMS) as reference (Supplementary Figs. 30–45). High-resolution matrix-assisted laser desorption ionization-time of flight mass spectrometry (MALDI-TOF MS) was performed on the Shimadzu spectrometer. (Supplementary Figs. 46–53).

### Device fabrication

The OSCs with ITO/PEDOT: PSS/ PM6:SMAs/PDINN/Ag structure were fabricated by consisting of the following five steps: (1) Pre-cleaning ITO-coated glass in turn with detergent, deionized water, acetone and isopropyl alcohol in an ultrasonic bath, then drying and treating ultraviolet ozone generator for 15 min. (2) Spinning coated PEDOT: PSS at 4000 rpm for 30 s and then annealing at 150 °C for 20 min. (3) Spinning coated the active layer (PM6:SMAs = 1:1.2; 16 mg/mL(CF); 0.5% (v/v) 1-CN), then annealing at 100 °C for 10 min in the glovebox. (4) Spinning coated the electron transport material PDINN. (5) Vaporizing 100 nm electrode (Ag) in a thermal evaporation chamber with a vacuum of -1 × 10$^{-4}$ Pa, obtained the OSCs with an effective area of 0.06 mm$^2$.

### J−V and EQE measurements

The current density−voltage (J−V) curve of OSCs was measured by a Keithley 2450 Source Measure Unit and an AAA class solar simulator (Model, Newport 94023 A) with 450 W xenon lamp under AM 1.5 G illumination. External quantum efficiency (EQE) was measured by the solar cell spectral response measurement system QE-R3011 (Taiwan Enli Technology Co., Ltd.).

### FTPS-EQE and EQE$_{EL}$

Fourier-transform photocurrent spectroscopy external quantum efficiency (FTPS-EQE) and external electroluminescence quantum efficiency (EQE$_{EL}$) of the optimized devices were measured by PECT-600 and ELCT3010 (Enlitech), respectively.

### The calculation processes of E$_{loss}$

The $E_{loss}$ is mainly composed of the following three components, the calculation formula is as follows:

1. Radiative recombination above the bandgap ($\Delta E_1$) (5)

$$\Delta E_1 = E_g - q V_{OC}^{SQ} \tag{7}$$

$$V_{OC}^{SQ} = \frac{kT}{q} \ln \left( \frac{q \int_0^\infty EQE_{PV}(E) \varnothing_{AM1.5}(E) dE}{q \int_{E_g}^\infty \varnothing_{BB}(E) dE} + 1 \right) \tag{8}$$

$$\varnothing_{BB}(E) = \frac{2\pi}{h^3 c^2} E^2 e^{-\frac{E}{kT}} \tag{9}$$

2. Radiative recombination blew the bandgap ($\Delta E_2$)

$$\Delta E_2 = E_{loss,rad} = q V_{OC}^{SQ} - q V_{oc}^{rad} \tag{10}$$

$$V_{oc}^{rad} = \frac{kT}{q} \ln\left(\frac{J_{SC}}{J_0^{rad}} + 1\right) = \frac{kT}{q} \ln\left(\frac{q\int_0^\infty EQE_{PV}(E)\,\varnothing_{AM1.5}(E)\,dE}{q\int_{E_g}^\infty \varnothing_{BB}(E)\,dE} + 1\right)$$

(11)

3. Non-radiative recombination loss ($\Delta E_3$)

$$\Delta E_3 = E_{loss,non-rad} = -\frac{kT}{q}\ln EQE_{EL}$$

(12)

$$\Delta E_3^{cal} = E_g - qV_{OC} - \Delta E_1 - \Delta E_2$$

(13)

where $E_g$, $V_{OC}^{SQ}$, $k$, $T$, $q$, $\varnothing_{BB}$, and $V_{oc}^{rad}$ refer to energy bandgap, Shockley-Queisser (SQ) $V_{OC}$ limit, the Boltzmann constant, the temperature, the elementary charge, the black body spectrum, and radiative recombination $V_{OC}$ limit, respectively.

### Reporting summary
Further information on research design is available in the Nature Portfolio Reporting Summary linked to this article.

### Data availability
The data supporting the findings of this study are available within the published article, Supplementary Information, Source Data Files, and Supplementary Data files. Additional data are available from the corresponding author on request. The X-ray crystallographic coordinates for structures reported in this study have been deposited at the Cambridge Crystallographic Data Centre (CCDC), under deposition numbers 2405808; 2405811; 2405812. Source data are provided with this paper.

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

## Acknowledgements

This work was supported by the Advanced Talents Incubation Program of the Hebei University (No. 521100224204 (J.G.)), the National Natural Science Foundation of China (Nos. 52403224 (J.G.), 52103235 (B.Q.), 52203248 (X.L.) and 52103243 (J.Z.)), the Beijing Nova Program (20240484597 (X.L.)) and the Strategic Priority Research Program of the Chinese Academy of Sciences (No. XDB 0520102 (L.M.)). A portion of this work is based on the data obtained at BSRF-1W1A. The authors gratefully acknowledge the cooperation of the beamline scientists at the BSRF-1W1A beamline.

## Author contributions

J.G. and B.Q. conceived the study. J.G synthesized and characterized acceptor materials and B.Q. performed device optimization and performance characterization. S.Q. carried out DFT calculation and data analysis. J.Z. conducted the TA measurements and data analysis. C.Z. carried out $E_{loss}$ measurements. H.Z. and Y.L. (Yuechen Li). conducted in UV-vis and CV characterization. X.X. performed GISAXS measurements. Y.G. participated in the synthesis of acceptors. T. L., Y. Z., G. H., and Y. Y. analyzed the single-crystal data. All authors contributed to discussions and commented on the manuscript. Y.L. (Yongfang Li), L.M., and J.C. directed the project. The manuscript was prepared, revised, and finalized by J.G., X.L., B.Q., and Y.L. (Yongfang Li).

## Competing interests

The authors declare no competing interests.

## Additional information

[1]Province-Ministry Co-construction Collaborative Innovation Center of Hebei Photovoltaic Technology, Hebei Key Laboratory of Optic-electronic Information and Materials, College of Physics Science and Technology, Hebei University, Baoding, Hebei, China. [2]Beijing National Laboratory for Molecular Sciences, CAS Key Laboratory of Organic Solids, Institute of Chemistry Chinese Academy of Sciences, Beijing, China. [3]National Engineering Research Center for Colloidal Materials, Shandong University, Jinan, Shandong, China. [4]Center for Physicochemical Analysis and Measurement, Institute of Chemistry, Chinese Academy of Sciences, Beijing, China. [5]Key Laboratory of Solid-State Optoelectronic Devices of Zhejiang Province, College of Physics and Electronic Information Engineering, Zhejiang Normal University, Jinhua, Zhejiang, China. [6]Laboratory of Advanced Optoelectronic Materials, Suzhou Key Laboratory of Novel Semiconductor-optoelectronics Materials and Devices, College of Chemistry, Chemical Engineering and Materials Science, Soochow University, Suzhou, Jiangsu, China. ✉e-mail: guojing@hbu.edu.cn; chenjianhui@hbu.edu.cn; lixiaojun@iccas.ac.cn; qiubeibei@zjnu.edu.cn

