## [Peer Review file · Nature Communications]

Asymmetric Small-Molecule Acceptor Enables Suppressed Electron-Vibration Coupling and Minimized Driving Force for Organic Solar Cells

Corresponding Author: Dr Beibei Qiu

Version 0:

Reviewer comments:

Reviewer #1

(Remarks to the Author)

In this manuscript, Guo et al. designed and synthesized a series of asymmetric acceptor molecules constructed from asymmetric side-chain groups (the combination of alkyl and thiophene alkyl chains), and proposed that they have significant advantages in reducing reorganization energy and obtaining minimum excitation energy, thereby reducing the non-radiative energy loss of the devices. Both theoretical and experimental results show that the combination of one-dimensional and two-dimensional asymmetric side-chains can reduce the non-radiative energy loss and improve the charge transport characteristics by affecting the strong electron-phonon coupling in acceptor molecules, and finally the OSCs based on BTP-BO-TBO achieve the highest PCE. This finding provides a unique insight into the inhibition of non-radiative energy loss in organic photovoltaic technology. Therefore, I highly recommend that this manuscript be accepted for publication in Nature Communications after addressing the following questions.

1. In Fig.1, the absorption spectrum curve was normalized for the UV measurement of the acceptor molecule. To better understand the optical absorption differences between different acceptor molecules, it is recommended to provide the acceptor molecular absorption coefficient.
2. Given that the results of TA are very interesting and hold significant implications for comprehending the working mechanism of organic photovoltaics, it is recommended to combine 2D transient absorption spectra for detailed discussion.
3. The energy loss analysis in this manuscript is a very important content, especially the energy value of the CT state is crucial for determining the minimum excited state ΔE_{LE-CT} . Many studies have shown that the energy of the CT state is very close to the energy of the LE state, please explain how the energy of the CT state is obtained. In addition, the fitting process of PM6: BTP-DC11 and PM6: BTP-C11-TBO systems should also be provided.
4. While the device preparation process was described in the "Method", the author did not give relevant data on the device optimization process. The related data also need to be provided.
5. The author adopted side-chain engineering to suppress non-radiative energy loss with the precondition of high charge generation efficiency and eventually achieved an inspiring PCE. However, stability studies are important for OSCs, have the authors conducted relevant studies and provided the results?
6. Some minor problems should be corrected and please carefully check the manuscript to prevent other errors, such as tense inconsistencies in the text on Page 5, line 22, "were shown" and "are displayed", and "are listed". such as there is no space in " $J_{SC} \propto P_{light}$ " should be " $J_{SC} \propto P_{light}$ " on Page 21, line 8.

Reviewer #2

(Remarks to the Author)

The authors provided an insight into the influence of asymmetric molecular structure on device performance of organic solar cells. The results may inspire molecular structure design of small molecule acceptors. While the article provides many simulation and experimental results to support the discussion, it should be revised on the following issues before it is considered for publication.

1. Advances in asymmetric side chains should be detailed to show the novelty of this work.
2. The authors claimed that "PM6:BTP-BO-TBO increases slowly to the maximum value and then decreases slowly, indicating charge separation and recombination are limited by exciton diffusion, so a higher FF is obtained." Could the

exciton lifetime be affected by the asymmetric structure?

3. A direct comparison of the photoluminescence and electroluminescence spectra of various systems is suggested to demonstrate the effect of asymmetric structure on nonradiative recombination.
4. Considering that the driving forces derived from Fig.4 b-d are very small and quite different from the TD-DFT calculation results, could the authors provide any fitting parameters (e.g., mean square error) to demonstrate that the results are not ruled by the fitting error?
5. For TA spectra, why does the signal at around 800 nm drop off faster than the signal at 633 nm?
6. Please clarify the aggregation behavior difference between various systems, and its influence on absorption edge, ΔE_{nr} , and E_{loss} .
7. Could the authors evaluate the contribution of electron-vibration coupling to E_{loss} ? The importance of electron-vibration coupling has not been fully expressed.

Reviewer #3

(Remarks to the Author)

In this manuscript written by Guo J et al, asymmetric NFAs have been studied and compared with symmetric NFAs from photophysical aspects. The relationship between vibrational coupling and non-radiative energy loss has been presented using theoretical and experimental measurement. Comprehensive characterizations have been carried out and the reason behind higher PCEs of asymmetric NFA based OSCs have been satisfactorily explained. The manuscript is well-written. However, I suggest authors to address following comments before I recommend the publication of the manuscript in Nature Communication.

Minor comments/questions for authors to address:

1. Introducing luminescent low-bandgap materials has a positive effect, especially on the Voc of device. However, in the introduction section, what does author meant by "a lot uncertainty in terms of film morphology" when introducing luminescent materials in the blend film of an OSC?
2. Recently, Jiang Y et al (<https://doi.org/10.1038/s41560-024-01557-z>) stated that in their Z8 asymmetric NFA, the presence of delocalized singlet exciton in Z8-NFA is the reason for lower CT-mediated recombinations via the LE-DSE-CS pathways and already published 20.2% (19.8% certified) PCE. So, what does author meant by "lack of comprehensive understanding regarding the mechanisms through which SMAs featuring asymmetric side-chains mitigate energy loss and charge-transport channels" in the introduction part? I suggest authors to add more detail about how to measure the energy of the hybrid CT-LE states.
3. Why does the sharp rise in the NFAs ESA at 990 nm is unfavorable for FF improvement in BTP-C11-TBO, while slow rise in the peak for BTP-BO-TBO is good for FF? More explanation is needed.
4. What authors can say about the relationship between electronic coupling and the molecular packing of these blend-films? If pi-pi stacking distance is reduced in asymmetric NFA (BTP-BO-TBO) based blend film, and more ordered structure is obtained, then, how does it affect electronic coupling? Does it increase or stay the same?
5. One of the assumptions in the manuscript is "the formation of relatively larger purer phase which prolonged the exciton diffusion mediated process", as described by τ_2 . Have authors measured the domain purity? If authors have measured domain purity, then, this assumption can be supported with experimental data. Otherwise, authors should provide more experimental supporting data here. If measurement of domain purity is time-sensitive and/or has less significance here in the context, then, authors should provide indirect measurement of domain purity or discuss domain purity with respect to exciton diffusion time in more detail, with support of more references in the literature.
6. How the SCLC region has been determined for the measurement of hole and electron mobilities? I suggest updating Figure S8 and clearly mentioning the SCLC region. Also, in the SCLC's method section in SI file, it would be better to clearly state the value of built-in voltage for clarity.
7. TEM images in Fig. 6 do not look like fibrous structures as the width is on the order of ~100-150 nm. These bright regions correspond to NFA as it is the bright-field TEM analysis where materials with high electron-affinity show dark regions. So, how to distinguish the effect of, especially, BTP-BO-TBO NFA based blend film with highest PCE?
8. The light green colors in the Figures (such as in Fig. 4b-d, and so on, is hard to clearly see. I suggest authors kindly update the light green color for better visibility. The values mentioned in the inset of Figures are too small to read. I suggest authors to update inset values in the Figure/Figures.

Version 1:

Reviewer comments:

Reviewer #1

(Remarks to the Author)

The authors well address the comments from the reviewers, I am pleased to recommend this revised version acceptable for publicaition in Nature Communications.

Reviewer #2

(Remarks to the Author)

Suggest to be published.

Reviewer #3

(Remarks to the Author)

Authors have addressed all of my concerns and thoroughly revised the manuscript. I suggest the acceptance of the manuscript in its current form.

Reviewer#1 (Remarks to the Author)

In this manuscript, Guo et al. designed and synthesized a series of asymmetric acceptor molecules constructed from asymmetric side-chain groups (the combination of alkyl and thiophene alkyl chains), and proposed that they have significant advantages in reducing reorganization energy and obtaining minimum excitation energy, thereby reducing the non-radiative energy loss of the devices. Both theoretical and experimental results show that the combination of one-dimensional and two-dimensional asymmetric side-chains can reduce the non-radiative energy loss and improve the charge transport characteristics by affecting the strong electron-phonon coupling in acceptor molecules, and finally the OSCs based on BTP-BO-TBO achieve the highest PCE. This finding provides a unique insight into the inhibition of non-radiative energy loss in organic photovoltaic technology. Therefore, I highly recommend that this manuscript be accepted for publication in the journal Nature Communications after addressing the following questions.

1. In Fig.1, the absorption spectrum curve was normalized for the UV measurement of the acceptor molecule. To better understand the optical absorption differences between different acceptor molecules, it is recommended to provide the acceptor molecular absorption coefficient.

Response: Thank you for your suggestions. To better understand the optical absorption differences between these acceptor molecules, the absorption coefficients of the acceptors have been added in Supplementary Fig 1 and Supplementary Table 1 in the revised Supplementary Information.

And we have added some explanations **on page 6-7** in the revised manuscript as follows: “In CF solution, the five SMAs exhibit nearly identical absorption spectra, with the coincident maximum absorption wavelength (λ_{\max}) located at about 733 nm, **and the absorption coefficients of them are**

measured to be in the range of $2.34 \sim 2.61 \times 10^5 \text{ M cm}^{-1}$ (Supplementary Fig 1 and Supplementary Table 1), indicating that the side chain structures of SMAs exert minimal influence on the intramolecular charge transfer absorption. In film state, both symmetric and asymmetric SMAs exhibit a similar trend, manifesting notable redshifts when compared with their corresponding solution absorptions (94 nm for BTP-DC11, 77 nm for BTP-DTBO, 68 nm for BTP-DBO, 89 nm for BTP-C11-TBO, 73 nm for BTP-BO-TBO, respectively). In addition, the asymmetric SMAs of BTP-C11-TBO ($1.02 \times 10^5 \text{ cm}^{-1}$) and BTP-BO-TBO ($1.07 \times 10^5 \text{ cm}^{-1}$) present the higher absorption coefficients than those of the symmetric SMAs (BTP-DC11, BTP-DTBO and BTP-DBO), suggesting the distinct aggregation behavior of these five SMAs in solid state.”

Supplementary Figure 1. a) Absorption spectra of BTP-DC11, BTP-DTBO, BTP-DBO, BTP-C11-TBO and BTP-BO-TBO in dilute chloroform solution; b) Absorption spectra of BTP-DC11, BTP-DTBO, BTP-DBO, BTP-C11-TBO and BTP-BO-TBO in solid state.

Supplementary Table 1. Absorption coefficient of SMAs (BTP-DC11, BTP-DTBO, BTP-DBO, BTP-C11-TBO and BTP-BO-TBO) in dilute chloroform solution and solid state.

Acceptor	Absorption coefficient	
	Solution $\times 10^5 [\text{M cm}^{-1}]$	Film $\times 10^5 [\text{cm}^{-1}]$
BTP-DC11	2.34	0.97
BTP-DTBO	2.48	0.96
BTP-DBO	2.39	0.97
BTP-C11-TBO	2.54	1.02
BTP-BO-TBO	2.61	1.07

2. Given that the results of TA are very interesting and hold significant implications for comprehending the working mechanism of organic photovoltaics, it is recommended to combine 2D transient absorption spectra for detailed discussion.

Response: We sincerely appreciate the valuable suggestions. In order to elucidate the distinctions between asymmetric and symmetrical acceptors more clearly, an in-depth analysis based on the results of the 2D transient absorption spectra have been added in the revised manuscript as follows: “To gain an in-depth understanding of the exciton dissociation and recombination dynamics, the BTP-DC11, BTP-DTBO, BTP-DBO, BTP-C11-TBO and BTP-BO-TBO based systems were investigated by femtosecond transient absorption (TAS) spectroscopy, as shown in Fig. 3. The 2D transient absorption spectra and the corresponding transient absorption spectra of these blend films at different decay times are displayed in Supplementary Fig. 6 and Supplementary Fig. 7 in the Supplementary Information, respectively. A pump wavelength of 800 nm was adopted to selectively excite the acceptors of the D:A blends. Fig. 3a and 3b compares the transient dynamics of the ground-state-bleach (GSB) signal of donor probing at 633 nm and excited state absorption (ESA) signal of acceptor probing at 990 nm in blend films, respectively. This donor GSB feature is an indication of ultrafast hole transfer from the SMAs exciton to PM6, producing the CT state at the donor-acceptor interface. For the acceptor ESA signal, in comparison with the symmetrical acceptor-based blends, the BTP-C11-TBO and BTP-BO-TBO based blends show relatively higher intensity of ESA signal centered at 900 nm after photoexcitation. Subsequently, as the ESA peak gradually decreased, a red-shifted ESA signal on a long timescale from 20 to 1000 ps could be observed, which should be assigned to the polaron states. The stronger and prolonged polaron states of the BTP-C11-TBO and BTP-BO-TBO based blend films indicated the lower bimolecular recombination probability in the corresponding devices. In addition, for the GSB signal of donor component, the kinetic curve of BTP-C11-TBO based blend reached the maximum value at the beginning and then rapidly decreased, suggesting the fast exciton separation and diffusion, which is conducive to obtaining a higher short-circuit current density (J_{sc}). Interestingly, BTP-BO-TBO based blend increases slowly to the maximum value and then decays slowly, demonstrating the longer exciton diffusion lifetime and CT state lifetime, which also indicates a significant reduction of charge recombination in the blend. In other words, the charge separation and recombination are limited by exciton diffusion, benefiting in the higher FF for PM6:BTP-BO-TBO based devices.”

Supplementary Figure 6. 2D transient absorption spectra of in PM6:BTP-DC11, PM6:BTP-DTBO, PM6:BTP-DBO and PM6:BTP-C11-TBO and PM6:BTP-BO-TBO blend films.

Supplementary Figure 7. Femtosecond transient absorption spectra of in PM6:BTP-DC11, PM6:BTP-DTBO, PM6:BTP-DBO and PM6:BTP-C11-TBO and PM6:BTP-BO-TBO blend films at selected time delays.

3. The energy loss analysis in this manuscript is a very important content, especially the energy value of the CT state is crucial for determining the minimum excited state ΔE_{LE-CT} . Many studies have shown

that the energy of the CT state is very close to the energy of the LE state, please explain how the energy of the CT state is obtained. In addition, the fitting process of PM6: BTP-DC11 and PM6: BTP-C11-TBO systems should also be provided.

Response: Thank you for your suggestions. Generally, the E_{CT} could be obtained by fitting the curves of s-EQE and EL, and the E_{LE} could be obtained through the intersection of s-EQE and EL curves. As the accurate measurement of the E_{LE} and E_{CT} values are of great importance to obtain the ΔE_{LE-CT} values, which reflects the strength of hybridization of E_{LE} and E_{CT} states, multiple fitting tests have been added and performed. The corresponding fitted results, including the calculated average values and standard deviation are shown in Supplementary Figure 16, and the re-calculated E_{loss} values have been updated in Table 3 in the revised manuscript. It could be noted that both of the calculated E_{CT} and ΔE_{LE-CT} values based on the five acceptor materials remain essentially unchanged, indicating the credibility of these calculated data. In addition, the fitting process of PM6: BTP-DC11 and PM6: BTP-C11-TBO based systems have added in the Supplementary Figure 16 in the revised Supplementary Information.

Supplementary Fig. 16 The s-EQE and EL curves of the devices based on PM6:BTP-DC11; PM6:BTP-DTBO; PM6:BTP-DBO; PM6:BTP-C11-TBO and PM6:BTP-BO-TBO.

Supplementary Table 9. Multi-fitting data statistics of s-EQE and EL curves for calculating ΔE_{LE-CT} .

Active layer	E_{LE} [eV]	E_{CT} [eV]	ΔE_{LE-CT} [eV]
PM6:BTP-DC11	1.37	1.34 (1.346±0.0115) ^a	0.03 (0.0233±0.0115) ^a
PM6:BTP-DTBO	1.43	1.38 (1.3775±0.005)	0.05 (0.0525±0.005)
PM6:BTP-DBO	1.45	1.42 (1.426±0.0115)	0.03 (0.0233±0.0115)
PM6:BTP-C11-TBO	1.39	1.36 (1.363±0.0057)	0.03 (0.0267±0.0057)
PM6:BTP-BO-TBO	1.40	1.38 (1.385±0.0058)	0.02 (0.0150±0.0058)

^{a)}The values in parentheses are the average values with standard deviations obtained from over 3 fits.

4. While the device preparation process was described in the “Method”, the author did not give relevant data on the device optimization process. The related data also need to be provided.

Response: Thank you for your suggestions. We have added data about the equipment optimization process to the supporting information, as detailed in Supplementary Table 6 and Supplementary Table 7.

Supplementary Table 6. Photovoltaic parameters of PM6:SMAs at different conditions with 100 °C, 10 min.

Active layer	Ratio	V_{oc} [V]	J_{sc} [mA cm ⁻²]	FF [%]	PCE [%]
PM6:BTP-DC11	1:1	0.854	26.58	78.13	17.73
	1:1.2	0.852	26.50	78.69	17.76
	1:1.5	0.852	26.10	78.39	17.43
PM6:BTP-DTBO	1:1	0.880	25.77	75.25	17.06
	1:1.2	0.881	25.64	76.23	17.22
	1:1.5	0.882	25.59	75.95	17.14
PM6:BTP-DBO	1:1	0.911	25.61	78.33	18.27
	1:1.2	0.909	25.89	78.27	18.42
	1:1.5	0.907	25.75	78.03	18.22
PM6:BTP-C11-TBO	1:1	0.858	27.32	78.67	18.44
	1:1.2	0.856	27.35	79.06	18.51
	1:1.5	0.854	27.25	78.53	18.28
PM6:BTP-BO-TBO	1:1	0.914	26.63	80.26	19.53
	1:1.2	0.913	26.67	81.17	19.76
	1:1.5	0.907	26.42	80.12	19.20

Supplementary Table 7. Photovoltaic parameters of PM6:SMAs at different conditions.

Active layer	Condition	V_{oc} [V]	J_{sc} [mA cm ⁻²]	FF [%]	PCE [%]
PM6:BTP-DC11	90 °C	0.854	26.33	78.38	17.62

	100 °C	0.852	26.50	78.69	17.76
	110 °C	0.851	26.59	78.27	17.71
	90 °C	0.883	25.45	76.01	17.08
PM6:BTP-DTBO	100 °C	0.881	25.64	76.23	17.22
	110 °C	0.879	25.52	75.81	17.01
	90 °C	0.910	25.67	77.84	18.18
PM6:BTP-DBO	100 °C	0.909	25.89	78.27	18.42
	110 °C	0.905	25.85	77.67	18.17
	90 °C	0.859	27.11	78.50	18.28
PM6:BTP-C11-TBO	100 °C	0.856	27.35	79.06	18.51
	110 °C	0.851	27.37	77.52	18.06
	90 °C	0.915	26.31	80.71	19.43
PM6:BTP-BO-TBO	100 °C	0.913	26.67	81.17	19.76
	110 °C	0.909	26.25	80.24	19.15

5. The author adopted side-chain engineering to suppress non-radiative energy loss with the precondition of high charge generation efficiency and eventually achieved an inspiring PCE. However, stability studies are important for OSCs, have the authors conducted relevant studies and provided the results?

Response: Thank you for your suggestions. The device stability, including maximum power point tracking stability, photostability and storage stability, of the asymmetric acceptor based OSCs (PM6: BTP-C11-TBO and PM6: BTP-BO-TBO) have been added in the revised Supplementary Information.

In addition, we added some descriptions in the revised manuscript as follows: “The operational stability of the two asymmetric acceptor molecules (PM6:BTP-C11-TBO and PM6:BTP-BO-TBO) based OSCs have been measured. Both of them retained above 95% initial efficiency after 600 s maximum power point tracking under continuous AM 1.5G illumination. In addition, after storage in nitrogen-filled glove box for 1600 h, both BTP-C11-TBO and PM6:BTP-BO-TBO based OSCs devices retained more than 80% of their initial PCEs. Furthermore, upon continuous white LED irradiation for 250 h, both of the devices maintained more than 90% of their initial PCEs. These results demonstrate the superior device stability of BTP-C11-TBO and PM6:BTP-BO-TBO based OSCs (as shown in Supplementary Fig 11 and Supplementary Fig 12 in the Supplementary Information).”

Supplementary Figure 11. The maximum power point tracking stability of OSCs based on the PM6: BTP-C11-TBO and PM6: BTP-BO-TBO devices.

Supplementary Figure 12. The operational stability of OSCs based on the PM6: BTP-C11-TBO and

PM6: BTP-BO-TBO devices: photostability (a-b) and storage stability (c-d).

6. Some minor problems should be corrected and please carefully check the manuscript to prevent other errors, such as tense inconsistencies in the text on Page 5, line 22, “were shown” and “are displayed”, and “are listed”. such as there is no space in “ $J_{sc} \propto P_{light}^a$ ” should be “ $J_{sc} \propto P_{light}^a$ ” on Page 21, line 8.

Response: Thank you for your careful check. We have checked the manuscript carefully, and have corrected these typos and errors in the revised manuscript. For example, in the text on page 5, line 22, both “are displayed” and “are listed” have been corrected to “were displayed” and “were listed”. On page 21, line 8, a space has been added to “ $J_{sc} \propto P_{light}^a$ ” changed to be “ $J_{sc} \propto P_{light}^a$ ”. In addition, similar errors in the manuscript have been thoroughly checked.

Reviewer #2 (Remarks to the Author):

The authors provided an insight into the influence of asymmetric molecular structure on device performance of organic solar cells. The results may inspire molecular structure design of small molecule acceptors. While the article provides many simulation and experimental results to support the discussion, it should be revised on the following issues before it is considered for publication.

1. Advances in asymmetric side chains should be detailed to show the novelty of this work.

Response: Thanks to the reviewer's suggestion. Most of the studies on asymmetric side-chain acceptor materials in the literature are in the length, substitution position, branched position and heteroatom doping of 1D alkyl side chains. As we all know, the 1D alkyl side chain has a significant role in improving molecular solubility and regulating molecular packing, while the 2D thiophene side chain has significant advantages in broadening the molecular absorption spectrum and reducing energy loss. Therefore, our work innovatively combined the 1D alkyl chain and the 2D thiophene side chain, hoping to achieve the effect of “1+1>2”, and deeply explores the effects of such asymmetric acceptor materials on molecular aggregation, molecular stacking and photovoltaic performance, which opens up new ideas for the study of asymmetric acceptor molecules. To highlight the innovative nature of our work, we have rewritten the relevant description in the introduction, focusing on the progress of research on asymmetric side-chain receptor molecules and the novelty of our work, on page 5 as show: “Inspired by their specific electronic structure and physical properties, modulating the chemical structures of A-DA’ D-A type SMAs have been extensively utilized by regulating the three

fundamental building blocks, including the central fused ring core unit (DA' D), electron-withdrawing end unit (A), and solubilizing side chains. Owing to the unique electron delocalization characteristics, the singlet exciton (SE) excited state of Y-series acceptors could be transferred into the intra-moiety excited (i-EX) state (or namely delocalized singlet exciton (DSE)), which is beneficial for charge generation and decreased non-radiative energy loss^{20, 21}. Modifications of the side chains, such as the variation of length, topology (linear or branched), branching points, and dimension are the most frequently employed approaches for precisely modulating the solubility, molecular crystallization, and stacking behavior of SMAs²²⁻²⁴. For instance, Yan et al. discovered that different positions of alkyl-chain-branching can alter the molecular stacking of SMAs, optimizing phase separation and exciton dissociation²⁵. It has been well established that the molecular stacking behavior can be regulated with improved structural order and charge transport in thin films by replacing the linear n-undecyl chain on Y6 with branched 2-butyloctyl, delivering significantly promoted open-circuit voltage (V_{oc}) and fill factor (FF)²⁶. Additionally, breaking the symmetry of the alkyl chains is also an effective way to boost photovoltaic performance^{20, 27}. For example, Yang and coworkers developed the hybrid cycloalkyl-alkyl chain-based symmetric/asymmetric acceptors Y-C10ch/A-C10ch, and the PM6:A-C10ch device based on asymmetric molecules had less energy loss²⁸. Development the acceptors with asymmetric side-chains, and an in-depth understanding of the intrinsic properties of their molecules, such as molecular packing, electron coupling, and charge transport properties, especially the electron vibrational coupling, which describes the deformations of the molecular geometries in the course of the electron-transfer process and reflects the interactions between electrons and intramolecular vibrations. Reducing the electron-vibrational coupling has been demonstrated as an effective strategy to suppress the non-radiative recombination, thus leading to suppressed non-radiative energy loss²⁹. Therefore, a comprehensive investigation of the molecular performance of side-chain asymmetric acceptors in combination with detailed density functional theory (DFT) horizontal calculations and single-crystal structures is essential for a deeper understanding of the energy loss and charge transport mechanisms of their devices.”

2. The authors claimed that “PM6:BTP-BO-TBO increases slowly to the maximum value and then decreases slowly, indicating charge separation and recombination are limited by exciton diffusion, so a higher FF is obtained.” Could the exciton lifetime be affected by the asymmetric structure?

Response: Thank you for your comments. Generally, the exciton lifetime is closely related with the chemical structures and the molecular packing features of the materials. To more clearly reveal the exciton lifetimes of different acceptor materials, the transient fluorescence measurements were

conducted on the thin pristine films of these acceptor materials. The detailed results are presented in Figure S9 and Table S7 in the revised Supplementary Information. It can be obtained that all the acceptors showed similar exciton lifetime with similar t_1 (50 ps) and t_2 (300 ps) values. However, as demonstrated by the results of film morphology and contact angle measurements, the PM6:BTP-BO-TBO based blend possessed the better molecular stacking features and higher phase purity, which might result in the longer exciton diffusion length for the BTP-BO-TBO component in the PM6:BTP-BO-TBO blend.

In addition, we have added some discussions on the exciton lifetime features in the revised manuscript as follows: “Furthermore, to get in-depth understanding on the exciton lifetimes of these acceptor materials, transient fluorescence measurements were tested to obtain the exciton lifetime of the pristine films. By fitting the attenuation curve with a double exponential (as shown in Supplementary Fig. 8 and Supplementary Table 3 in the Supplementary Information), it can be found that all the acceptors showed similar t_1 (50 ps) and t_2 (300 ps) values, suggesting the similar exciton lifetime of these thin acceptor films. Besides, considering that the exciton lifetime is also closely related with the molecular packing features, we suspect that the BTP-BO-TBO component in the PM6:BTP-BO-TBO blend possess the longer exciton diffusion length, thus results in the longer exciton diffusion lifetime.”

Supplementary Figure 8. Streak camera images of the photoluminescence of a) BTP-DC11, b) BTP-DTBO, c) BTP-DBO, d) BTP-C11-TBO and e) BTP-BO-TBO pristine acceptor films; f) Photoluminescence decay traces of the pristine acceptor films.

Supplementary Table 3. Double-exponential fitting results of photoluminescence decay traces of the original acceptor.

Film	t_1 (ps)	t_2 (ps)
BTP-DC11	52.17 ± 1.98	330.22 ± 10.8
BTP-DTBO	55.97 ± 2.04	300.37 ± 8.52
BTP-DBO	49.09 ± 1.38	309.75 ± 8.74
BTP-C11-TBO	53.05 ± 2.11	316.60 ± 9.37
BTP-BO-TBO	51.12 ± 1.51	316.70 ± 9.23

3. A direct comparison of the photoluminescence and electroluminescence spectra of various systems is suggested to demonstrate the effect of asymmetric structure on nonradiative recombination.

Response: We sincerely appreciate the valuable suggestions. Generally, the higher PL fluorescence quantum yield (PLQY) and EQE_{EL} value suggest the smaller non-radiative energy loss (ΔE_{nr}) (*Nat Commun.* **14**, 5079 (2023); *Nat Commun.* **13**, 3256 (2022)). To demonstrate the effect of asymmetric structures on non-radiative recombination, the photoluminescence and electroluminescence spectra of these acceptor-based systems have been added. And the corresponding results have been presented in Supplementary Figure 2 and Supplementary Table 2 in the revised Supplementary Information. It could be found that among these acceptor-based systems, the BTP-BO-TBO based system exhibited the highest PL quantum yield and EQE_{EL} value, which is consistent with the corresponding photovoltaic performance.

In addition, we have added some discussions about the photoluminescence in the revised manuscript as follows: “In both solutions and films, the stokes shifts of BTP-BO-TBO is smaller than that of other SMAs, suggesting that the excited state relaxation in BTP-BO-TBO is smaller, which is beneficial for the associated voltage losses. This is in agreement with the reduced reorganization energy for the transition between the ground state and the first excited state (S1). In addition, compared with the solutions, the films exhibit relatively larger stokes shifts (Supplementary Fig. 2). This is presumably due to the fact that there exists an energy disorder for the S1 state in the films, and the excitons on the molecules with higher S1 energy can transfer to the molecules with lower S1 energy to emit photons³¹. Generally speaking, the higher the PL fluorescence quantum yield (PLQY), the smaller the nonradiative energy loss, and the results in Supplementary Table 2 show that BTP-C11-TBO and BTP-BO-TBO have the highest PLQY in both solution and thin film, which also fully confirms the above results³².”

Besides, some discussions about the electroluminescence have also been added in the revised manuscript as follows: “Moreover, the electroluminescence quantum efficiencies (EQE_{EL}) of the

devices are shown in Fig. 5g, and the E_{QEEL} values of the five devices are measured to be 1.84×10^{-4} for PM6:BTP-DC11, 1.22×10^{-4} for PM6:BTP-DTBO, 2.27×10^{-4} for PM6:BTP-DBO, 2.51×10^{-4} for PM6:BTP-C11-TBO and 4.75×10^{-4} for PM6:BTP-BO-TBO, respectively, among which the E_{QEEL} of the PM6:BTP-BO-TBO device is higher than those of the other devices. According to the equation: $\Delta E_{nr} = -\frac{KT}{q} \ln(EQE_{EL})$, the ΔE_{nr} of these devices could be obtained, as presented in Table 3. Ultimately, the devices constructed by PM6:BTP-BO-TBO achieved the lowest ΔE_{nr} of 0.198 eV and the smaller E_{loss} values of 0.485 eV.”

Supplementary Figure 2. a) Normalized PL spectra and Stokes shifts of five acceptors in chloroform solution. b) Normalized PL spectra and Stokes shifts of five acceptors in films.

Supplementary Table 2. PL quantum yield measured in solution and film.

Material	Solution	Film
	Φ_{PL} (%)	Φ_{PL} (%)
BTP-DC11	13.49	6.25
BTP-DTBO	8.60	2.62
BTP-DBO	11.04	5.33
BTP-C11-TBO	14.40	6.88
BTP-BO-TBO	15.42	7.02

Fig. 5| g. EQEEL curves of the optimal OSCs.

Table 3. Total energy losses and detailed energy losses of the optimized devices.

Active layer	E_g [eV]	V_{OC} [V]	E_{loss} [eV]	ΔE_1 [eV]	ΔE_2 [eV]	ΔE_3 [eV]	EQEEL [$\times 10^{-4}$]
PM6:BTP-DC11	1.366	0.851	0.515	0.257	0.035	0.223	1.84
PM6:BTP-DTBO	1.428	0.878	0.550	0.262	0.055	0.233	1.22
PM6:BTP-DBO	1.457	0.900	0.557	0.265	0.073	0.219	2.27
PM6:BTP-C11-TBO	1.374	0.856	0.518	0.259	0.044	0.215	2.51
PM6:BTP-BO-TBO	1.392	0.907	0.485	0.260	0.027	0.198	4.75

4. Considering that the driving forces derived from Fig.4 b-d are very small and quite different from the TD-DFT calculation results, could the authors provide any fitting parameters (e.g., mean square error) to demonstrate that the results are not ruled by the fitting error?

Response: We sincerely appreciate the valuable comments. As the accurate measurement of the E_{LE} and E_{CT} values are of great importance to obtain the ΔE_{LE-CT} values, multiple fitting tests have been added and performed. The corresponding fitted results have been added as Supplementary Figure 16, and the calculated average values and standard deviation data have been summarized in Supplementary Table 9 in the revised Supplementary Information. Besides, the re-calculated E_{loss} values have been updated in Table 3 in the revised manuscript. It could be noted that, both of the calculated E_{CT} and ΔE_{LE-CT} values for the five acceptor-based devices remain essentially unchanged, indicating that the

experimental results are basically not affected by the fitting error, and the data are credible.

Supplementary Figure 16. The s-EQE and EL curves of the devices based on PM6:BTP-DC11; PM6:BTP-DTBO; PM6:BTP-DBO; PM6:BTP-C11-TBO and PM6:BTP-BO-TBO.

Supplementary Table 9. Multi-fitting data statistics of s-EQE and EL curves for calculating ΔE_{LE-CT} .

Active layer	E_{LE} [eV]	E_{CT} [eV]	ΔE_{LE-CT} [eV]
PM6:BTP-DC11	1.37	1.34 (1.346±0.0115) ^a	0.03 (0.0233±0.0115) ^a
PM6:BTP-DTBO	1.43	1.38 (1.3775±0.005)	0.05 (0.0525±0.005)
PM6:BTP-DBO	1.45	1.42 (1.426±0.0115)	0.03 (0.0233±0.0115)
PM6:BTP-C11-TBO	1.39	1.36 (1.363±0.0057)	0.03 (0.0267±0.0057)
PM6:BTP-BO-TBO	1.40	1.38 (1.385±0.0058)	0.02 (0.0150±0.0058)

^{a)} The values in parentheses are the average values with standard deviations obtained from over 3 fits.

5. For TA spectra, why does the signal at around 800 nm drop off faster than the signal at 633 nm?

Response: We sincerely appreciate the valuable comments. According to the UV-Vis absorption profiles, the signal at 633 nm should be ascribed to the GSB signal of polymer donor, and the signal at around 800 nm should be ascribed to the GSB signal of SMA component. In this manuscript, all the five low-bandgap acceptor materials exhibit wide absorption profiles ranging from 300 nm to ~1000 nm, and the wide-bandgap polymer donor possess a narrower absorption profile ranging from 300 nm

to ~700 nm. In order to better study the hole transfer process without the interference of electron or energy transfer, an excitation wavelength of 800 nm was adopted to selectively excite the acceptor component of the D:A blends. Thus, the GSB signal of SMA component (~800 nm) was generated by the direct excitation by the pump laser. However, the GSB signal of polymer donor was generated by the hole transfer from the acceptor component to the donor component at the donor-acceptor interface, due to the HOMO energy level offset between the polymer donor and SMAs. In brief, the evolution mechanism of the GSB signal of SMA component (~800 nm) is different from the GSB signal of polymer donor (~633 nm). The faster drop off in the early stage for the signal at around 800 nm should be ascribed to the additional decay channel of hole transfer from the acceptor component to the donor component.

6. Please clarify the aggregation behavior difference between various systems, and its influence on absorption edge, ΔE_{nr} , and E_{loss} .

Response: Thank you for your suggestions. To gain a better understanding of the aggregation behavior differences in these acceptor-based systems, single crystal structures of three acceptors (symmetric acceptors BTP-DBO and BTP-DTBO, and asymmetric acceptor BTP-BO-TBO) and STEM images at different scales have been measured, demonstrating the significant influence of the side chain structures on the molecular packing and phase separation behaviors, as shown in Supplementary Fig. 19 and Supplementary Fig. 20. The single crystal results indicate that the introduction of thiophene side chains can promote the formation of more charge transport channels between molecules, and the introduction of asymmetric side chains leads to the formation of tight π - π packing of the end-group overall (average π - π stacking distance (\bar{d}): $\bar{d}_{BTP-BO-TBO}(3.3125 \text{ \AA}) < \bar{d}_{BTP-DTBO}(3.3295 \text{ \AA})$). And the STEM results suggest that the film morphologies of asymmetric acceptor-based blend films are mainly dominated by the aliphatic alkyl chains, and the thiophene alkyl chain mainly functions as a morphology regulator.

Besides, to better evaluate the influence of the aggregation behavior on the absorption edge, ΔE_{nr} , and E_{loss} , the normalized PL spectra and Stokes shifts of the five acceptor materials in dilute solution and film state have been added. It could be found that all the five SMAs exhibited nearly identical absorption spectra. However, in film state, these SMAs exhibited notable but different redshifts, resulting in the different optical bandgaps of 1.359, 1.348, 1.379, 1.336 and 1.375 eV for BTP-DC11, BTP-DTBO and BTP-DBO, BTP-C11-TBO and BTP-BO-TBO, respectively, which should be ascribed to the distinct aggregation behavior of these five SMAs in solid state. Furthermore, different stokes shifts could also be obtained for these acceptor films. Among the five SMA films, the BTP-BO-

TBO film presented the smallest stokes shift, which is in agreement with the reduced reorganization energy for the transition between the ground state and the first excited state (S1), benefiting in obtaining reduced non-radiative energy losses.

In addition, we have added some discussions about the influence of the aggregation behavior in the revised manuscript as follows: “In CF solution, the five SMAs exhibit nearly identical absorption spectra, with the coincident maximum absorption wavelength (λ_{\max}) located at about 733 nm, indicating that the side chain structures of SMAs exert minimal influence on the intramolecular charge transfer absorption. In film state, both symmetric and asymmetric SMAs exhibit a similar trend, manifesting notable redshifts than their corresponding solution absorptions (94 nm for BTP-DC11, 77 nm for BTP-DTBO, 68 nm for BTP-DBO, 89 nm for BTP-C11-TBO, 73 nm for BTP-BO-TBO, respectively), which should be ascribed to the distinct aggregation behavior of these five SMAs in solid state. Meanwhile, according to their absorption edge onsets (the intersection of the tangent line of the absorption peak edge and the vertical axis $Y = 0$), the optical bandgaps of BTP-DC11, BTP-DTBO and BTP-DBO, BTP-C11-TBO and BTP-BO-TBO are estimated to be 1.359, 1.348, 1.379, 1.336 and 1.375 eV, respectively. In both solutions and films, the stokes shifts of BTP-BO-TBO is smaller than that of other SMAs, suggesting that the excited state relaxation in BTP-BO-TBO is smaller, which is beneficial for the associated voltage losses. This is in agreement with the reduced reorganization energy for the transition between the ground state and the first excited state (S1). In addition, compared with the solutions, the films exhibit relatively larger stokes shifts (Supplementary Fig. 2). This is presumably due to the fact that there exists an energy disorder for the S1 state in the films, and the excitons on the molecules with higher S1 energy can transfer to the molecules with lower S1 energy to emit photons³¹. Generally speaking, the higher the PL fluorescence quantum yield (PLQY), the smaller the nonradiative energy loss, and the results in Supplementary Table 2 show that BTP-C11-TBO and BTP-BO-TBO have the highest PLQY in both solution and thin film, which also fully confirms the above results³².” (*Nat. Commun.*, 2022,13, 3256; *Nat. Commun.*, 2023, 14, 5079)

Supplementary Figure 2. a) Normalized PL spectra and Stokes shifts of five acceptors in chloroform solution. b) Normalized PL spectra and Stokes shifts of five acceptors in films.

Supplementary Figure 19. The STEM phase images (1 μm , 500 nm and 200 nm scale) of the blend films of PM6:SMAs.

Supplementary Figure 20. The single crystal and the π - π stacking in one unit cell and the 3D network packing of BTP-DBO, BTP-DTBO and BTP-BO-TBO along the c-axis.

7. Could the authors evaluate the contribution of electron-vibration coupling to E_{loss} ? The importance of electron-vibration coupling has not been fully expressed.

Response: Thank you for your suggestions. Generally, electron-vibrational coupling describes the deformations of the molecular geometries in the course of the electron-transfer process and reflects the interactions between electrons and intramolecular vibrations. Strong electron-vibrational coupling signifies that a substantial amount of energy is allocated to vibrational modes during the photoelectric conversion processes, such as exciton decay, charge-transfer (CT) state decay and nongeminate (bimolecular) recombination, leading to increased energy dissipation and elevated reorganization energy. Reducing the electron-vibrational coupling has been demonstrated as an effective strategy to suppress the non-radiative recombination, thus leading to suppressed non-radiative energy loss (*Nat. Energy* **2017**, 2, 17053., *Nat. Commun.* **2022**, 13, 3256.).

Specifically, the detailed components of V_{loss} in OSCs could be categorized into three parts based on the Shockley-Queisser (SQ) limit, as shown in Equation (1):

$$\begin{aligned}
qV_{loss} &= E_g - qV_{OC} \\
&= (E_g - qV_{OC}^{SQ}) + (qV_{OC}^{SQ} - qV_{OC}^{rad}) + (qV_{OC}^{rad} - qV_{OC}) \\
&= (E_g - qV_{OC}^{SQ}) + q\Delta V_{OC}^{rad, belowgap} + q\Delta V_{OC}^{non-rad} \\
&= \Delta E_1 + \Delta E_2 + \Delta E_3
\end{aligned} \tag{1}$$

The first two parts (ΔE_1 and ΔE_2) in Equation (1) are caused by radiative recombination, and the third part (ΔE_3) is caused by nonradiative recombination. The energy loss due to non-radiative recombination $q\Delta V_{OC}^{non-rad}$ can be rewritten as:

$$\Delta E_3 = q\Delta V_{OC}^{non-rad} = \Delta E_{nr} = -k_B T \ln(EQE_{EL}) = -k_B T \ln\left(\frac{k_r}{k_r + k_{nr}}\right) \tag{2}$$

where EQE_{EL} represents radiative quantum efficiency of the OSC when charge carriers are injected into the device in dark, and the EQE_{EL} could be defined as the ratio of the rate constant of radiative decay from the CT state to the ground state (k_r) and the total decay-rate constant, which is the sum of k_r and the nonradiative recombination-rate constant k_{nr} .

The nonradiative rate constant (k_{nr}) can be described, by the electron transfer-rate constant expression using Fermi's golden rule and the Born-Oppenheimer approximation, as a product of the electronic coupling V between the CT state and ground state and the Franck-Condon weighted density of states $FCWD(\hbar\omega)$ (*Physical Review X* **2018**, 8, 031055.):

$$k_{nr} = \frac{2\pi}{\hbar} V^2 FCWD(0) \tag{3}$$

The Franck-Condon weighted density of states accounts for transitions between all vibrational modes of the initial (CT) state and all vibrational modes of the final (ground) state where the states differ in energy by $\hbar\omega$. For nonradiative decay, it takes the argument 0. In general,

$$FCWD(\hbar\omega) = \frac{1}{\sqrt{4\pi\lambda_l k_B T}} \sum_{w=0}^{\infty} \sum_{t=0}^{\infty} \frac{e^{-S} S^{w-t} t!}{w!} [L_t^{w-t}(S)]^2 e^{-\{[\hbar\omega - \Delta G_0 + \lambda_l + (w-t)\hbar\Omega]^2 / 4\lambda_l k_B T\}} e^{-t\hbar\Omega/k_B T} \tag{4}$$

where λ_l represents the low-frequency reorganization energy, $S = (\lambda_v / \hbar\Omega)$ is the Huang Rhys factor, and λ_v and Ω are the reorganization energy and the harmonic frequency of the quantized high-frequency mode, respectively (assumed the same for all vibronic states). ΔG_0 is the difference in Gibbs free energy between the two states. w and t designate the quantum number of the vibrational modes of the ground state and the CT state, respectively, and $L_t^{w-t}(S)$ is the generalized Laguerre polynomial of degree t .

Based on these equations mentioned above, it could be concluded that the suppressed electron-vibration coupling could result in the decreased Franck-Condon weighted density of states $FCWD(\hbar\omega)$, thus leading to the reduced nonradiative rate constant (k_{nr}). Therefore, reduced non-radiative energy loss and V_{loss} could be realized.

Typically, in terms of the charge-transfer process, according to the classical Marcus electron-transfer theory:

$$k_{ET} = V_{if}^2 \sqrt{\frac{\pi}{\lambda k_B T \hbar^2}} \exp\left[-\frac{(\Delta G + \lambda)^2}{4\lambda k_B T}\right] \quad (5)$$

where λ is the reorganization energy, V represents the electronic coupling between the initial state and the final state, ΔG is the free energy change. A small reorganization energy can facilitate the reduction of the driving force required for exciton dissociation. Therefore, the reorganization energy plays a crucial role in the photoelectric conversion and the energy loss processes for OSCs.

In this manuscript, based on the results of DFT calculations and photoluminescence experiments, it could be obtained that the asymmetric acceptors exhibited the smaller reorganization energy during photoelectric conversions, demonstrating that asymmetric side chain strategy could effectively suppress the electron-vibration coupling. Besides, the two asymmetric acceptors showed the higher PL quantum yields, and the devices based on the two asymmetric acceptors also presented the higher EQE_{EL} values. Especially, the BTP-BD-TBO based device delivered the highest EQE_{EL} of 4.75×10^{-4} , leading to the lowest $\Delta V_{OC}^{non-rad}$ of 0.198 V. In summary, these results demonstrate the importance of suppressing the electron-vibration coupling in reducing the energy loss of OSCs.

In addition, we have added some discussions to highlight the contribution of electron-vibration coupling to E_{loss} in the revised manuscript as follows: “Additionally, breaking the symmetry of the alkyl chains is also an effective way to boost photovoltaic performance^{20, 27}. For example, Yang and coworkers developed the hybrid cycloalkyl-alkyl chain-based symmetric/asymmetric acceptors Y-C10ch/A-C10ch, and the PM6:A-C10ch device based on asymmetric molecules had less energy loss²⁸. Development the acceptors with asymmetric side-chains, and an in-depth understanding of the intrinsic properties of their molecules, such as molecular packing, electron coupling, and charge transport properties, especially the electron vibrational coupling, which describes the deformations of the molecular geometries in the course of the electron-transfer process and reflects the interactions between electrons and intramolecular vibrations. Reducing the electron-vibrational coupling has been demonstrated as an effective strategy to suppress the non-radiative recombination, thus leading to suppressed non-radiative energy loss²⁹. Therefore, a comprehensive investigation of the molecular performance of side-chain asymmetric acceptors in combination with detailed density functional theory (DFT) horizontal calculations and single-crystal structures is essential for a deeper understanding of the energy loss and charge transport mechanisms of their devices.”

Reviewer #3 (Remarks to the Author):

In this manuscript written by Guo J et al, asymmetric NFAs have been studied and compared with symmetric NFAs from photophysical aspects. The relationship between vibrational coupling and non-radiative energy loss has been presented using theoretical and experimental measurement. Comprehensive characterizations have been carried out and the reason behind higher PCEs of asymmetric NFA based OSCs have been satisfactorily explained. The manuscript is well-written. However, I suggest authors to address following comments before I recommend the publication of the manuscript in Nature Communication.

Minor comments/questions for authors to address:

1. Introducing luminescent low-bandgap materials has a positive effect, especially on the Voc of device. However, in the introduction section, what does author meant by “a lot uncertainty in terms of film morphology” when introducing luminescent materials in the blend film of an OSC?

Response: We sincerely appreciate the valuable comments. Generally, the luminescent characteristics of the materials exhibit a significant correlation with the non-radiative energy losses in devices. Enhancing the luminescence efficiency of low-bandgap materials has been demonstrated to be a crucial strategy for reducing non-radiative energy losses and improving the open-circuit voltage (V_{oc}) of the corresponding organic solar cells (OSCs). In the “Introduction Section”, the statement of “a lot uncertainty in terms of film morphology” by “introducing luminescent moieties” was intended to highlight the importance of the film morphology in determining the photovoltaic performance, as the morphology-related issues could result in a reduced fill factor (FF), thereby impairing the overall efficiency of the devices (*Adv. Energy Mater.* 2021, 11, 2102596; *Energy Environ. Sci.*, 2021,14, 3469-3479). Therefore, synergistically improving the luminescent properties of materials and optimizing the morphological structure of the active layer have been become the key to further improve the photovoltaic properties.

In addition, to enhance the clarity and precision of the statement, the related statement in the “Introduction Section” have been modified in the revised manuscript as follows: “... Recent studies have demonstrated that introducing luminescent moieties such as fused ring groups (thiophene, benzene, etc.) to enhance the rigidity of molecular structures and/or the extent of electron delocalization could be an effective method for achieving highly luminescent donor and acceptor molecules, as well as optimal photovoltaic performance^{18, 19}. In addition, as the film morphology is closely related with the charge transport properties, and thus the FF and overall efficiency of OSCs. Synergistically improving the luminescence efficiencies of materials and optimizing the morphological structure of the active layer have been become the key to further improve the photovoltaic properties

but with great challenges....”

2. Recently, Jiang Y et al (<https://doi.org/10.1038/s41560-024-01557-z>) stated that in their Z8 asymmetric NFA, the presence of delocalized singlet exciton in Z8-NFA is the reason for lower CT-mediated recombinations via the LE-DSE-CS pathways and already published 20.2% (19.8% certified) PCE. So, what does author meant by “lack of comprehensive understanding regarding the mechanisms through which SMAs featuring asymmetric side-chains mitigate energy loss and charge-transport channels” in the introduction part? I suggest authors to add more detail about how to measure the energy of the hybrid CT-LE states.

Response: Thank you for your suggestions. Recent studies have found that owing to the unique electron delocalization characteristics resulting from its distinctive molecular structure and packing behavior of Y6, the singlet exciton (SE) excited state at approximately 900 nm of Y6 could be transferred into the intra-moiety excited (i-EX) state (or namely delocalized singlet exciton (DSE)) at approximately 1580 nm, resulting in the formation of DSE (*J. Am. Chem. Soc.* **2020**, *142*, 12751.). The formation of DSE has been demonstrated to be beneficial for charge generation and for reducing non-radiative energy loss, which elucidates the excellent photovoltaic properties of Y6. Generally, as the Y-series acceptors usually possess similar chemical structures and packing characteristics with Y6 to some extent, recent studies have shown that the DSE was also formed in these Y-series acceptor-based systems (*Nat. Commun.* **2024**, *15*, 3287., *Energy Environ. Sci.* **2023**, *16*, 3373., *Angew. Chem. Int. Ed.* **2024**, doi:10.1002/anie.202415994. and *Nat. Energy* **2024**, *9*, 975.). In this manuscript, owing to the limited detection range of the transient absorption measurement (~ 500-1100 nm), we were unable to directly observe and study these potential DSE signals.

In addition, to enhance the clarity and precision of the statement, the related statement “lack of comprehensive understanding regarding the mechanisms through which SMAs featuring asymmetric side-chains mitigate energy loss and charge-transport channels” in the “Introduction Section” have been modified in the revised manuscript as follows: “... **Owing to the unique electron delocalization characteristics, the singlet exciton (SE) excited state of Y-series acceptors could be transferred into the intra-moiety excited (i-EX) state (or namely delocalized singlet exciton (DSE)), which is beneficial for charge generation and decreased non-radiative energy loss^{20, 21}.**”

In terms of the hybridization of the LE and CT states, recent studies have demonstrated that enhancing the hybridization of the LE and CT states is conducive to suppressing the non-radiative recombination, thereby reducing the non-radiative voltage loss (ΔV_{nr}) of the OSC devices (*Nat. Energy* **2021**, *6*, 799., *Adv. Energy Mater.* **2023**, *13*, 2301026.). However, the method for accurately measuring

the energy level of the hybrid LE-CT states have not been reported, according to recent studies. We can only determine the degree of hybridization of the LE and CT states through specific experiments, and investigate the effects of hybrid LE-CT state with different degrees of hybridization on the photovoltaic properties of OSCs. Generally, the strength of the hybridization of LE and CT state is closely related with the energy offset between LE and CT states (ΔE_{LE-CT}). The smaller ΔE_{LE-CT} indicate the stronger hybridization of LE and CT states. In this manuscript, to accurately evaluate the effects of asymmetric molecular structures on the hybridization of LE and CT states, TD-DFT calculation on the D:A complex configuration, and s-EQE and EL measurement on the acceptor based OSCs have been applied to determine the values of E_{LE} , E_{CT} and ΔE_{LE-CT} , as shown in Fig. 2, Supplementary Fig. 17 and Supplementary Table 10. These results demonstrated that the asymmetric acceptor BTP-BO-TBO-based system possess the smaller ΔE_{LE-CT} value, in comparison with the symmetric acceptors, suggesting the stronger hybridization between the CT and LE states, which thus contributes to the smaller ΔE_{nr} of PM6:BTP-BO-TBO-based devices.

In addition, and we have added more descriptions of the mixed CT-LE state in the revised manuscript as follows: “Recent studies demonstrated that enhancing the hybridization of the LE and CT states is conducive to suppressing the non-radiative recombination, thereby reducing the non-radiative voltage loss (ΔV_{nr}) of the OSC devices. As the strength of the hybridization of LE and CT state is closely related with the energy offset between LE and CT states (ΔE_{LE-CT}). The smaller ΔE_{LE-CT} indicate the stronger hybridization of LE and CT states. Therefore, a combination of theoretical and experimental approach was applied to accurately evaluate the effects of asymmetric molecular structures on the hybridization of LE and CT states.”

3. Why does the sharp rise in the NFAs ESA at 990 nm is unfavorable for FF improvement in BTP-C11-TBO, while slow rise in the peak for BTP-BO-TBO is good for FF? More explanation is needed.

Response: We sincerely appreciate the valuable comments. In this manuscript, both BTP-C11-TBO and BTP-BO-TBO based blends showed fast rise ESA signals at 990 nm upon excitation. We surmise that the reviewer might refer to the kinetics of the GSB signal at 633 nm. This relatively slower evolution should be attributed to the interfacial dissociation of the excitons that migrate from the inside of the acceptor domains to the interfaces, which reflects the exciton diffusion in the acceptor domains. As shown in Fig. S6 and Table S1 in the Supplementary Information, by fitting the hole transfer kinetics with a bi-exponential function, BTP-BO-TBO based blends exhibited the relatively smaller τ_1 of 0.49 ps but a larger τ_2 of 12 ps, when compared with BTP-C11-TBO based films (τ_1 of 0.57 ps and τ_2 of 3.8 ps). The longer τ_2 value suggest the relatively longer exciton diffusion lifetime of BTP-BO-

TBO based blend films. According to the equation: $L_D = (D\tau)^{1/2}$, where D and τ represent diffusion coefficient and exciton lifetime, respectively. The longer exciton diffusion lifetime of BTP-BO-TBO based blend films indicate the longer exciton diffusion length, which might be ascribed to the better molecular stacking features, enhanced phase separation, and higher phase purity, as demonstrated by the results of film morphology analysis and contact angle measurements. In brief, the longer exciton diffusion lifetime indicates the larger exciton diffusion length of BTP-BO-TBO based blend films, which could contribute to the improved charge transport and reduced charge recombination, and thus the enhanced FF of BTP-BO-TBO based devices.

In addition, we have added some discussions in the revised manuscript as follows: “A pump wavelength of 800 nm was adopted to selectively excite the acceptors of the D:A blends. Fig. 3a and 3b compares the transient dynamics of the ground-state-bleach (GSB) signal of donor probing at 633 nm and excited state absorption (ESA) signal of acceptor probing at 990 nm in blend films, respectively. This donor GSB feature is an indication of ultrafast hole transfer from the SMAs exciton to PM6, producing the CT state at the donor-acceptor interface. For the acceptor ESA signal, in comparison with the symmetrical acceptor-based blends, the BTP-C11-TBO and BTP-BO-TBO based blends show relatively higher intensity of ESA signal centered at 900 nm after photoexcitation. Subsequently, as the ESA peak gradually decreased, a red-shifted ESA signal on a long timescale from 20 to 1000 ps could be observed, which should be assigned to the polaron states. The stronger and prolonged polaron states of the BTP-C11-TBO and BTP-BO-TBO based blend films indicated the lower bimolecular recombination probability in the corresponding devices. In addition, for the GSB signal of donor component, the kinetic curve of BTP-C11-TBO based blend reached the maximum value at the beginning and then rapidly decreased, suggesting the fast exciton separation and diffusion, which is conducive to obtaining a higher short-circuit current density (J_{sc}). Interestingly, BTP-BO-TBO based blend increases slowly to the maximum value and then decays slowly, demonstrating the longer exciton diffusion lifetime and CT state lifetime, which also indicates a significant reduction of charge recombination in the blend. In other words, the charge separation and recombination are limited by exciton diffusion, benefiting in the higher FF for PM6:BTP-BO-TBO based devices.”

4. What authors can say about the relationship between electronic coupling and the molecular packing of these blend-films? If pi-pi stacking distance is reduced in asymmetric NFA (BTP-BO-TBO) based blend film, and more ordered structure is obtained, then, how does it affect electronic coupling? Does it increase or stay the same?

Response: Thank you for your suggestions. To gain a better understanding of the relationship between the molecule structures, the electronic coupling results and the molecular packing behaviors of these

acceptors with or without asymmetric side chain structures, the single crystal structures of three acceptors (symmetric acceptors of BTP-DBO and BTP-DTBO, and asymmetric acceptor of BTP-BO-TBO) and the corresponding electronic coupling values have been added and calculated, as shown in Supplementary Table 10 and Supplementary Fig. 20 and 21 in the revised Supplementary Information. It is noteworthy that all three acceptors exhibited 3D network packing structures, but with distinct features and varying π - π stacking distances, indicating the significant influence of the side chain structure on the molecular packing behavior. Besides, similar results could also be obtained from the calculated electronic coupling ($|J|$) results between a reference molecule and its nearest molecule in a single crystal structure.

In addition, we have added some discussions about the electronic couplings in the revised manuscript as follows: “To fully understand the difference in the stacking behavior of SMAs with side-chain symmetric and asymmetric, we first explored the single-crystal structure^{54, 55}. Compared with BTP-DBO, BTP-DBO and BTP-BO-TBO both have four different stacking modes in a single unit cell, providing a richer transport channel for charge transfer. Meanwhile, the single crystal structure showed that the Mode 2: A/A and Mode 3:D-A/A-D in BTP-BO-TBO have a shorter π - π distance than BTP-DTBO (Supplementary Fig. 20), which indicates that the introduction of asymmetric side chains can promote formation tighter π - π packing. Subsequently, we calculated electronic coupling ($|J|$) based on single crystal structure to estimate the charge transfer among SMAs²¹. As shown in Supplementary Fig. 21, taking 1 as a reference molecule, the adjacent molecules named 2, 3, 4, etc., respectively, the electron-coupled pairs formed are represented by 1-2, 1-3, 1-4..., etc. The BTP-DBO, BTP-DTBO and BTP-BO-TBO single crystal all possesses 4 nearest neighbors and 4 electron-coupled pairs, and the detailed $|J|$ are summarized in Supplementary Table 11. The relative order of the overall $|J|$ values of the three molecules is as follows: $|J|(\text{BTP-DTBO}) > |J|(\text{BTP-BO-TBO}) > |J|(\text{BTP-DBO})$. Significantly, however, the $|J|_{\text{max}}$ of BTP-BO-TBO is located 1-2 dimer (Mode 1:D-A/A-D), rather than the 1-4 dimer (Mode 3:D-A/A-D) in BTP-DTBO and BTP-DBO single crystals. The former has a larger stacking area between molecules, which is more conducive to charge transfer⁵⁶.” (*Energy Environ. Sci.*, 2024,**17**, 6844-6855; *Angew. Chem. Int. Ed.* 2023, **62**, e202313016; *Energy Environ. Sci.*, 2022,**15**, 4601-4611; *Angew. Chem. Int. Ed.* 2021, **60**, 15348-15353)

Supplementary Figure 20. The single crystal and the π - π stacking in one unit cell and the 3D network packing of BTP-DBO, BTP-DTBO and BTP-BO-TBO along the c-axis.

Supplementary Table 10. Crystal data for BTP-DBO (CCDC 2405808), BTP-DTBO (CCDC 2405811) and BTP-BO-TBO (CCDC 2405812).

Acceptor	BTP-DBO	BTP-DTBO	BTP-BO-TBO
Space group	C2/c	C2/c	C2/c
a [Å]	27.0612(7)	27.5504(13)	28.0817(11)
b [Å]	22.3128(6)	57.0641(14)	56.7025(17)
c [Å]	32.2933(9)	13.5264(5)	13.5233(5)
alpha [°]	90	90	90
beta [°]	110.561(3)	93.431(4)	93.635(4)
gamma [°]	90	90	90
Volume [Å ³]	18260.3(9)	21227.3(14)	21489.9(13)

Supplementary Figure 21. a) Illustration of the nearest-neighbor molecular pairs and b) the corresponding electron transfer integrals calculated at the B3LYP/6-31G** level for BTP-DBO, BTP-DTBO, and BTP-BO-TBO (For clarity, the alkyl side chains are shortened by methyl groups and the hydrogen atoms are omitted).

Supplementary Table 11. Electron coupling of BTP-DBO, BTP-DTBO, and BTP-BO-TBO. Dimers refer to the numbering scheme in the figures in the main text.

Dimer	$ J /\text{meV}$		
	BTP-DBO	BTP-DTBO	BTP-BO-TBO
1-2	10.98	15.76	36.09
1-3	8.30	10.06	11.05
1-4	26.06	63.79	5.07
1-5	8.30	15.76	7.05

5. One of the assumptions in the manuscript is “the formation of relatively larger purer phase which prolonged the exciton diffusion mediated process”, as described by τ_2 . Have authors measured the domain purity? If authors have measured domain purity, then, this assumption can be supported with experimental data. Otherwise, authors should provide more experimental supporting data here. If

measurement of domain purity is time-sensitive and/or has less significance here in the context, then, authors should provide indirect measurement of domain purity or discuss domain purity with respect to exciton diffusion time in more detail, with support of more references in the literature.

Response: We sincerely appreciate the valuable comments. Generally, domain purity is a crucial morphological parameter that has a significant impact on charge separation and transport, and thus the FF of OSCs (*Chem. Rev.* **2012**, *112*, 5488., *Adv. Energy Mater.* **2017**, *7*, 1700084.). And the domain purity could be determined in a quantitative manner by resonant soft X-ray scattering (R-SoXS). However, due to the limited testing time of R-SoXS, we were unable to identify a research group capable of conducting this measurement. According to the recent studies, the Flory-Huggins interaction parameter (χ) of the donor and acceptor materials is closely related with the domain purity of the blend films (*Nat. Mater.* **2018**, *17*, 253.). Higher χ values of the donor and acceptor materials would lead to greater domain purity in the blend films. Therefore, the contact angle measurements have been added and conducted to calculate the χ values of the donor PM6 and the five SMAs, as shown in Supplementary Fig. 10. The corresponding χ values are presented in Supplementary Table 5. It could be found that the χ values of the donor PM6 and BTP-DC11, BTP-DTBO, BTP-DBO, BTP-C11-TBO and BTP-BO-TBO blends were calculated to be 0.28 K, 0.64 K, 0.49K, 0.62K and 0.74 K, respectively. Among them, the χ of PM6 and BTP-BO-TBO presented the highest value, indicating the better domain purity of the BTP-BO-TBO based blend film.

In addition, we have added some discussions about the domain purity in the revised manuscript as follow: "...Therefore, to evaluate the differences in domain purity among these blend films, the surface energies (SEs) and corresponding Flory-Huggins interaction parameter (χ) values of these materials were calculated by contact angle measurements, as shown in Supplementary Fig. 10 and Supplementary Table 5. The closer the SE means the better the miscibility between the donor and acceptor³⁹. According to the empirical formula $\chi = K(\sqrt{\gamma_D} - \sqrt{\gamma_A})^2$, where K is a constant, γ_D/γ_A represents the SE of the donor/acceptor, and the χ values of the donor PM6 and five SMAs blends were calculated to be 0.28 K, 0.64 K, 0.49K, 0.62K and 0.74 K, respectively. Among them, the χ of PM6: BTP-BO-TBO blend film presented the highest value, indicating the higher domain purity of the BTP-BO-TBO based blend film. As mentioned above, the asymmetric molecular geometry can promote exciton diffusion charge transfer while inhibiting charge recombination, thereby reducing the non-radiative voltage loss^{30, 40,41}.

BTP-DC11	BTP-DTBO	BTP-DBO	BTP-C11-TBO	BTP-BO-TBO	PM6
					H ₂ O CA=95.3°	H ₂ O CA=96.4°	H ₂ O CA=97.1°	H ₂ O CA=96.4°	H ₂ O CA=97.0°	H ₂ O CA=100.3°
					CH ₂ I ₂ CA=48.7°	CH ₂ I ₂ CA=43.4°	CH ₂ I ₂ CA=45.6°	CH ₂ I ₂ CA=43.6°	CH ₂ I ₂ CA=42.1°	CH ₂ I ₂ CA=59.3°

Supplementary Figure 10. The contact angle images of BTP-DC11, BTP-DTBO, BTP-DBO, BTP-C11-TBO and BTP-BO-TBO Film.

Supplementary Table 5. The contact angle and the corresponding surface energy of different functional films.

Film	H ₂ O [°]	CH ₂ I ₂ [°]	SE [mN m ⁻¹]	γ_d [mN/m]	γ^p [mN/m]	χ [a.u.]
BDP-DC11	95.3	48.7	35.12	35.09	0.03	0.28K
BDP-DTBO	96.4	43.4	38.34	38.32	0.03	0.64K
BTP-DBO	97.1	45.6	37.14	37.12	0.02	0.49K
BTP-C11-TBO	96.4	43.6	38.22	38.20	0.02	0.62K
BTP-BO-TBO	97.0	42.1	39.19	39.12	0.07	0.74K
PM6	100.3	59.3	29.10	29.08	0.01	\

$$\chi = K(\sqrt{\gamma_D} - \sqrt{\gamma_A})^2$$

6. How the SCLC region has been determined for the measurement of hole and electron mobilities? I suggest updating Figure S8 and clearly mentioning the SCLC region. Also, in the SCLC 's method section in SI file, it would be better to clearly state the value of built-in voltage for clarity.

Response: Thank you for your suggestions. The SCLC region for the measurement of charge mobility could be determined by the [$\log(J)$ vs. $\log(V)$] curves fitted at the slope around 2. The fitted [$\log(J)$ vs. $\log(V)$] curves for the clearly mentioning the SCLC region have been added as Figure S8 in the revised Supplementary Information. It can be obtained that the voltage ranging from 1 to 4 V belongs to the SCLC region. Thus, the carrier mobility could be calculated from the slope of the $J^{1/2} \sim V$ curves in the SCLC region.

In addition, the SCLC mobilities could be calculated by Mott-Gurney equation:

$$J(V) = \frac{9}{8} \epsilon_0 \epsilon_r \mu_0 \frac{V^2}{L^3}$$

Where $V = V_{applied} - V_{built-in} - V_{series-resistance}$ (the $V_{built-in}$ values are 0.2 V and 0 V for the hole-only and the electron-only devices, respectively), $V_{applied}$ is the voltage applied, and $V_{built-in}$ is the built-in voltage from the relative work function difference between the two electrodes, and $V_{series-resistance}$ is the voltage caused by the series and contact resistance potential drop. For convenience, the voltage drops caused by the $R_{series-resistance}$ was ignored.

Besides, the detailed information of SCLC measurement, including the value of built-in voltage, have been updated in the Methods section in the revised Supplementary Information as follows:

4. SCLC mobility measurements

Electron-only devices with the structure of ITO/ZnO/ active layer/PDINN/Ag and hole-only devices with the structure of ITO/ PEDOT: PSS/active layer/ MoO₃/Ag are used to conduct SCLC measurements. The mobilities were determined by fitting the dark-field current density-voltage curves using the Mott-Gurney relationship, which is described in the following equation:

$$J(V) = \frac{9}{8} \epsilon_0 \epsilon_r \mu_0 \frac{V^2}{L^3} \quad (1)$$

where J is the current density, ϵ_0 is the permittivity of free space, $\epsilon_r \approx 3.5$ is the average dielectric constant of the blend film, μ_0 is the zero-field mobility, $V = V_{applied} - V_{built-in} - V_{series-resistance}$ (the $V_{built-in}$ values are 0.2 V and 0 V for the hole-only and the electron-only devices, respectively), $V_{applied}$ is the voltage applied, and $V_{built-in}$ is the built-in voltage from the relative work function difference between the two electrodes, and $V_{series-resistance}$ is the voltage caused by the series and contact resistance potential drop. For convenience, the voltage drops caused by the $R_{series-resistance}$ was ignored, L is the thickness of the active layer. The SCLC region for the measurement of charge mobility could be determined by the $[\log(J) \text{ vs. } \log(V)]$ curves fitted at the slope around 2. Thus, the carrier mobilities could be calculated from the slope of the $J^{1/2} \sim V$ curves in the SCLC region.

Supplementary Figure 14. a) J - V curves of electron-only devices. b) J - V curves of hole-only devices.

7. TEM images in Fig. 6 do not look like fibrous structures as the width is on the order of ~ 100 - 150 nm. These bright regions correspond to NFA as it is the bright-field TEM analysis where materials with high electron-affinity show dark regions. So, how to distinguish the effect of, especially, BTP-BO-TBO NFA based blend film with highest PCE?

Response: We sincerely appreciate the valuable comments. Owing to the relatively low resolution of the TEM images, it is challenging to differentiate among the TEM images of these acceptor-based blend films in this scale. To more effectively distinguish the impacts of the side chain structures on the blend film morphology, the TEM images with larger scales and the STEM images with better resolution and greater clarity have been added as Fig. S11 and Fig. S12, respectively, in the revised Supplementary Information. It could be obtained that the TEM images of PM6: BTP-BO-TBO based blends showed a more dense and uniform black-and-white distribution in comparison with other blend films. Furthermore, the STEM results demonstrated that the PM6: BTP-BO-TBO based blend films exhibited delicate fibrous aggregations with a 3D interpenetrating network structure. This morphological feature is beneficial for the enhanced charge transport and reduced charge recombination, and thus resulting in the optimal photovoltaic performance of PM6: BTP-BO-TBO based OSCs.

In addition, we have added some discussions on the film morphology in the revised manuscript as follows: **“To acquire a profound understanding of the influence of various side chains on the film morphology of the active layers, atomic force microscopy (AFM), transmission electron microscopy (TEM) and scanning transmission electron microscopy (STEM) were conducted on these PM6:SMAs**

blend films. As depicted in Fig. 6a, compared with the blends based on BTP-DC11 and BTP-DBO, the root-mean-square roughness (RMS) of the blends based on BTP-C11-TBO (0.903 nm) and BTP-BO-TBO (0.868 nm) decreased after the introduction of thiophene side chains, especially that of PM6:BTP-BO-TBO, showing a more uniform and smoother surface characteristics. Moreover, the TEM images of PM6:BTP-BO-TBO showed a more dense and uniform black-and-white distribution compared with other blend films (Fig. 6b and Supplementary Fig. 18). It is worth noting too that the STEM images of PM6:BTP-BO-TBO exhibited a clearer and more pronounced phase separation structure characteristics of fibrous 3D network showed in Supplementary Fig. 19. Additionally, it is interesting to note that the STEM images of PM6:C11-TBO and PM6:BO-TBO are relatively consistent with those of PM6:BTP-DC11 and PM6:BTP-DBO, respectively, that is that PM6:C11-TBO and PM6:BTP-DC11 show cluster-like aggregation, while PM6:BO-TBO and PM6:DBO mainly exhibit fibrous-like aggregation, indicating that the morphology regulation of donor-acceptor blends based on side-chain asymmetric acceptor is mainly affected by aliphatic alkyl chains while the thiophene alkyl chain had little effect. The above results manifest that modifying the side chains of SMAs is conducive to achieving the preferable phase separation morphology of blend films, promoting charge transport and improving FF, thereby optimizing photovoltaic performance.”

Supplementary Figure 18. The TEM phase images (1 μm and 500 nm scale) of the blend films of PM6:SMAs.

Supplementary Figure 19. The STEM phase images (1 μm , 500 nm and 200 nm scale) of the blend films of PM6:SMAs.

8. The light green colors in the Figures (such as in Fig. 4b-d, and so on, is hard to clearly see. I suggest authors kindly update the light green color for better visibility. The values mentioned in the inset of Figures are too small to read. I suggest authors to update inset values in the Figure/Figures.

Response: We sincerely appreciate the valuable suggestions. In order to enhance the visibility of the Figures, the light green colors initially utilized in these Figures have been replaced with a more pronounced shade of blue in the revised manuscript. In addition, the font size of the values mentioned in the inset of these Figures have also been enlarged to achieve better clarity in the revised manuscript.

We appreciate your considerations and look forward to hearing from you.

Yours sincerely,

Beibei Qiu

College of Physics and Electronic Information Engineering

Zhejiang Normal University

Jinhua 321004 China

E-mail: qiubei@iccas.ac.cn